# SMALL VARIANCE, BIG FAIRNESS: A PATH TO HARMLESS FAIRNESS WITHOUT DEMOGRAPHICS

## ABSTRACT

Statistical fairness harnesses a classifier to accommodate parity requirements by equalizing the model utility (e.g., accuracy) across disadvantaged and advantaged groups. Due to privacy and security concerns, recently there has arisen a need for learning fair classifiers without ready-to-use demographic information. Existing studies remedy this challenge by introducing various side information about groups and many of them are found fair by unavoidably comprising model utility. *Can we improve fairness without demographics and without hurting model utility?* To address this problem, we propose to center on minimizing the variance of training losses, allowing the model to effectively eliminate possible accuracy disparities without knowledge of sensitive attributes. During optimization, we develop a dynamic harmless update approach operating at both loss and gradient levels, directing the model towards fair solutions while preserving its intact utility. Through extensive experiments across four benchmark datasets, our results consistently demonstrate that our method effectively reduces group accuracy disparities while maintaining comparable or even improved utility.

## 1 INTRODUCTION

Fairness in machine learning has gained significant attention owing to its multifaceted ethical implications and its far-reaching potential to shape and influence various aspects of society (Dwork et al., 2012; Barocas & Selbst, 2016; Ntoutsi et al., 2020). In high-stakes decision-making domains, algorithms that merely prioritize model utility may yield biased models, resulting in unintentional discriminatory outcomes concerning factors such as gender and race. Group fairness, a.k.a. statistical fairness (Carey & Wu, 2023), addresses this issue by explicitly encouraging the model behavior to be independent of group indicators, such as disparate impact (Feldman et al., 2015), or equalized odds (Hardt et al., 2016). However, with increasing privacy concerns applied in practical situations, sensitive attributes are not accessible which raises a new challenge for fairness learning.

According to literature, numerous efforts have been directed towards achieving fairness without relying on demographic information, which can be mainly categorized into two branches. One branch is to employ proxy-sensitive attributes (Yan et al., 2020; Grari et al., 2021; Zhao et al., 2022; Zhu et al., 2023). These works assume that estimated or selected attributes are correlated with the actual sensitive attributes and thus can serve as a proxy of potential biases. The other branch follows Rawlsian Max-Min fairness (Rawls, 2001) whose main idea is to enhance the utility of the worst-off group. Existing studies (Hashimoto et al., 2018; Martinez et al., 2021; Chai & Wang, 2022b) following this principle commonly leverage a prior about group size to identify the worst-group members, which has been however argued at a non-negligible cost on overall utility (Chai et al., 2022).

The trade-off nature between model utility and fairness has emerged as a subject of dynamic discourse (Dutta et al., 2020; Wei & Niethammer, 2022; Zhao & Gordon, 2022), even in the context of no demographics (Martinez et al., 2021). Instead of walking on the Pareto frontier of utility and fairness, in this work, we advocate how much we can improve fairness during training without hurting the model's utility, which is particularly preferred in utility-intensive scenarios (Li & Liu, 2022). We also emphasize that fairness here refers to group utility disparity (e.g., accuracy difference across groups) following John Rawls's theory on distributive justice while no additional prior about demographics is introduced.

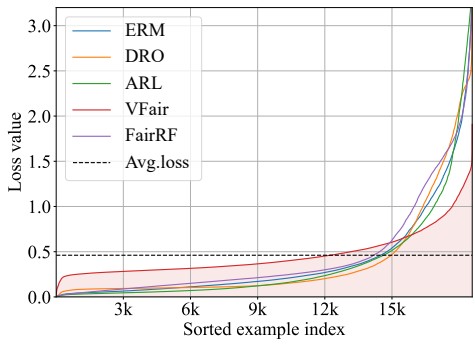

(a) Per-example training loss sorted in an ascending order.

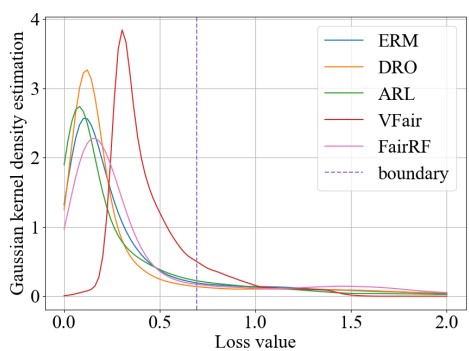

(b) Loss distribution with Gaussian kernel density estimation.

Figure 1: The effect of VFair by comparing with four methods on the training set of Law School.

*Our idea.* In this paper, we approach the problem from a novel perspective. The crux of the desired fairness lies in the pursuit of minimum group utility disparity across all groups. Since during the training phase, we are not aware of what the actual sensitive attributes are used for test data, the safest way is to ensure every possible disjoint group of training data has the same utility. To this end, we expect the training loss for each individual example to be approximately uniform, meaning that the variance of losses is sufficiently small. Considering such expectations may deteriorate the overall model performance, we prioritize keeping the overall average loss at a low value. Hence, the main idea can be summarized as minimizing loss distribution's second moment while not increasing its first moment, dubbed as VFair in this paper.

*Observation.* The implementation of VFair is presented in Section 3. Figure 1 here showcases the effect of our model VFair in contrast to Empirical Risk Minimization (ERM), DRO (Hashimoto et al., 2018), ARL (Lahoti et al., 2020), and FairRF (Zhao et al., 2022) on the training set of the Law School dataset (Wightman, 1998). To align with the requirement of harmless fairness, each trained model is kept with its average loss equal to ERM (with a maximal gap of 5e-3 in practical implementations). And we have the following findings.

(1) Our method VFair exhibits a more flattened curve compared to others while upholding a comparable area under the curve (filled with pink), signifying a harmless fairness solution. From Fig 1 (a), we observe that minimizing the variance of losses efficiently flattens the loss curve, showing a fairer solution for unknown group partitions on the test phase. Fig 1 (b) shows that the distribution mode (sharper than others) of our VFair slightly shifts towards a larger loss but keeps a distance from the decision boundary, which is an affordable cost demonstrated by its generalization in Section 4.2).

(2) Our method VFair experimentally implies the worst-case fairness, proved in Fig. 1 (a) that the average loss of the worst-off group for VFair will be consistently lower than any other method (Please also refer to the results on other datasets shown as Fig. 4 in Section 4.2). Given a prior about the minimal group size, the loss of worst-case fairness is defined as averaging the largest losses within this size. Our claim is obviously true if the group size is small, e.g., the worst-group examples are picked from 16k to end. Regarding a larger group size, e.g., picking training examples from 6k, thanks to the fact the total area under each curve is nearly equal and the curve of VFair is always above others before 16k, we conclude that the worst-group loss for VFair is also the lowest.

*Our contributions.* The primary contributions of our research can be outlined as follows.

• We highlight the setting of harmless fairness without prior knowledge about demographics. To well position this setting, we study its connection with existing literature from different views.

• We advocate that minimizing the variance of prediction losses is a straightforward yet effective fairness proxy. By incorporating it as a secondary objective, the overall model performance remains uncompromised.

• We develop a dynamic approach to do harmless update, which is operated at both the loss and gradient levels, comprehensively guiding the model towards a harmless fairness solution.

• We evaluate our framework across four benchmark datasets. The experimental results consistently demonstrate it rivals state-of-the-art methods.

## 2 RELATED WORK

**Max-Min fairness without demographics.** In alignment with Rawlsian Max-Min fairness (Rawls, 2001) principle, a sequence of studies has addressed the challenge of fairness without demographics by focusing on improving the performance of the worst-off group. DRO (Hashimoto et al., 2018) identified the worst-off group members with the assistance of a lower bound for the minimal group ratio. The behind insight is that examples yielding larger losses are more likely sampled from an underprivileged group and thus should be up-weighted, which inherits the fairness strategy for handling group imbalance (Abernethy et al., 2022; Chai & Wang, 2022a). Similarly, Martinez et al. (2021) also considered training a fair model with a given ratio of the protected group and connected such a fairness learning setting with the subgroup robustness problem (Liu et al., 2021). In contrast to these studies, ARL (Lahoti et al., 2020) presented an adversarial reweighing method for achieving fairness without resorting to any prior knowledge about demographic information. This embodies the genuine essence of achieving fairness without demographics and is closest to our setting.

**Harmless fairness.** In utility-intensive scenarios, a fair model is meaningful only when it preserves good utility. A few studies have engaged in discussing the extent to which fairness can be achieved without compromising model utility. Martinez et al. (2020; 2021) searched for the so-called minimax Pareto fair optimality for off-the-shelf binary attributes and then upgraded their method to the multi-value attribute cases with only side information about group size. Li & Liu (2022) accomplished cost-free fairness through a pre-processing strategy that involves reweighing training examples based on both fairness-related measures and predictive utility on a validation set. Based on the concept of Rashomon Effect, Coston et al. (2021) achieved fairness from good-utility models through a constrained optimization perspective, needing a proper upper bound for the average loss. Notably, these works more or less require direct or implicit demographic information and cannot adapt to our problem setting. Gong & Liu (2021) introduced a unified dynamic barrier gradient descent algorithm allowing models to prioritize must-satisfactory constraints. Inspired by this, we conceptualize harmless fairness within a similar framework, enabling us to move beyond a utility-only solution and obtain a fairer model which can narrow the utility gaps among possible data partitions.

## 3 VFAIR METHODOLOGY

### 3.1 PROBLEM SETUP

Consider a supervised classification problem from input space $\mathcal{X}$ to a label space $\mathcal{Y}$, with training set $\{z_i\}_{i=1}^N$, where $z_i = (x_i, y_i) \in \mathcal{X} \times \mathcal{Y}$. For a model parameterized by $\theta \in \Theta$ and a training point $z^1$, let $\ell(z; \theta)$ be the associated loss. In the context of fairness learning, we assume that for each $z_i$, a sensitive attribute $s_i \in \mathcal{S}$ exits, and a $K$-value sensitive attribute $s$ naturally partitions data into $K$ disjoint groups. We highlight such sensitive attributes are not observed during training but can be used for fairness testing. For example, let $a_k$ be the accuracy of the $k$-th group, and we can define the maximum accuracy disparity, i.e., MAD $= \max_{i,j \in [K]} (a_i - a_j)$, as a fairness metric. The fundamental objective of this work is to develop a classifier that maintains high overall predictive accuracy (compared to ERM) while minimizing group accuracy disparity (Lahoti et al., 2020; Martinez et al., 2021) to the greatest extent possible.

### 3.2 FAIRNESS VIA MINIMIZING VARIANCE OF LOSSES

An ERM model may exhibit variable predictive accuracy across different groups. Normally, a fair counterpart is achievable by properly incorporating the objective of minimizing group accuracy disparity (e.g., MAD), which is however not applicable when demographics are not accessible. It is worth noting that a classifier that can be fair for any possible partitions in testing implies that the loss of each training example should be close to each other. A compelling piece of evidence is that if all

---

[1]Throughout this paper, random variables are represented with lowercase letters unless otherwise specified.

training examples are correctly classified, each individual loss $\ell(z;\theta)$ should be small enough and no disparity should appear, i.e., MAD $= 0$. Although taking the risk of being overfitting, this case inspires us to bypass the unobserved sensitive attribute $s$ by requiring the loss values to be restricted in a relatively narrow range. One can move to Appendix A for a further justification.

As already mentioned in Section 1, we consider penalizing the variance of training losses to reduce the group accuracy disparity. Supposing that we intend to get a small MAD through minimizing the maximum group loss disparity, denoted by $\ell_{\text{MAD}}$, the following proposition shows that deviation $\mathbb{V}_z[\ell(z;\theta)]$ actually serves as a useful proxy.

**Proposition 1** $\forall s \in \mathcal{S}, \forall \theta \in \Theta, \ell_{MAD} \leqslant C\sqrt{\mathbb{V}_z[\ell(z;\theta)]}$, where $C$ is a constant.

Please refer to Appendix B.1 for the proof. Notably, although $\ell_{\text{MAD}}$ is upper-bounded in the form of standard deviation as stated in Proposition 1, we use variance for convenience in statements where it does not introduce ambiguity. Further discussions on the adoption of alternative loss options can be referred to Appendix B.2. Since penalizing the variance of loss will not necessarily decrease the expectation of losses $\mathbb{E}_z[\ell(z;\theta)]$, which could lead to the emergence of a uniform classifier (Martinez et al., 2021), we formulate the full objective as follows:

$$\min_{\theta \in \Theta} \sqrt{\mathbb{V}_z[\ell(z;\theta)]} \quad s.t. \ \mathbb{E}_z[\ell(z;\theta)] \leqslant \delta, \tag{1}$$

where $\delta$ controls how much we can tolerate the harm on the overall predictive accuracy, and the sense that $\delta = \inf_{\theta \in \Theta} \mathbb{E}_z[\ell(z;\theta)]$ suggests a zero-tolerance. In particular, we fold in any regularizers into $\ell(\cdot)$ to make our method easily adapt to specific scenarios. The empirical risk of Eq. 1 is written as

$$\min_{\theta \in \Theta} \underbrace{\sqrt{\frac{1}{N}\sum_{i=1}^{N}(\ell(z_i;\theta) - \hat{\mu}(\theta))^2}}_{\hat{\sigma}(\theta)} \quad s.t. \ \underbrace{\frac{1}{N}\sum_{i=1}^{N}\ell(z_i;\theta)}_{\hat{\mu}(\theta)} \leqslant \delta. \tag{2}$$

We use $\hat{\mu}(\theta)$ and $\hat{\sigma}(\theta)$ to denote the primary and secondary objectives, respectively. Since we minimize $\hat{\sigma}(\theta)$ inside the optimal set of minimizing $\hat{\mu}(\theta)$, the eventually learned model is viewed to be harmlessly fair with regard to the overall performance. Recall that Fig. 1 visualizes the effect of variance penalization and the training loss distribution of our VFair method.

## 3.3 HARMLESS UPDATE

Directly calculating the optimal set of $\hat{\mu}(\theta) \leqslant \delta$ in Eq. 2 can be very expensive. A common approach is to consider the unconstrained form of Eq. 2, i.e., Lagrangian function, which however needs to not only specify a proper $\delta$ beforehead but also optimize the Lagrange multiplier to best satisfy the constraint. Recognizing that such re-balancing between two loss terms essentially operates on gradients, in a manner analogous to the approach outlined by Gong & Liu (2021), we consider the following gradient update paradigm

$$\theta^{t+1} \leftarrow \theta^t - \eta^t\left(\lambda^t\nabla\hat{\mu}(\theta^t) + \nabla\hat{\sigma}(\theta^t)\right), \tag{3}$$

where $\eta^t$ is a step size and $\lambda^t(\lambda^t \geqslant 0)$ is the dynamic coefficient we aim to get. For simplicity, we omit the $t$ when it is not necessary to emphasize the step.

Our aim is to ensure the gradient updates in a direction that aligns with the primary objective's gradient without causing conflicts. As depicted in Fig. 2 (a), when the angle between gradients $\nabla\hat{\sigma}$ and $\nabla\hat{\mu}$ forms an obtuse angle, a detrimental component emerges in the direction of $\nabla\hat{\mu}$. Conversely, in Fig. 2 (b), the update will not conflict. Consequently, $\lambda$ should be sufficiently large to guarantee the combined force's component in the primary direction remains non-negative:

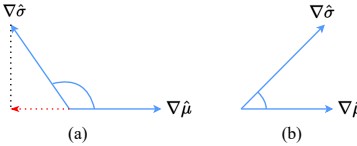

Figure 2: Two situations when updating primary and secondary gradient simultaneously.

$$\lambda\nabla\hat{\mu} + \text{Proj}_{\nabla\hat{\mu}}(\nabla\hat{\sigma}) \geqslant \epsilon\nabla\hat{\mu} \implies \lambda \geqslant \epsilon - \frac{\nabla\hat{\mu} \cdot \nabla\hat{\sigma}}{||\nabla\hat{\mu}||^2} := \lambda_1. \tag{4}$$

Here, $\epsilon$ represents the extent to which we wish to update $\nabla\hat{\mu}$ when it reaches a right angle. Thus, Eq. 4 can be simplified as $\lambda \geqslant \epsilon$. Throughout the experiments of this paper, we simply apply $\epsilon$ by checking its performance in the range of [-1,3] with a step size of 1.

The combined gradient on each instance is in the form of (detailed derivation is referred to Appendix C.)

$$\nabla = \lambda\nabla\hat{\mu} + \nabla\hat{\sigma} = \frac{1}{N}\sum_{i=1}^{N}\underbrace{\left(\lambda + \frac{1}{\hat{\sigma}}(\ell_i - \hat{\mu})\right)}_{w_i}\frac{\partial\ell_i}{\partial\theta}. \tag{5}$$

From Eq. 5, we observe that our objective (i.e., Eq. 1) implicitly leads to an example re-weighing scheme. As we can see that $w_i < 0$ if $\ell_i < \hat{\mu} - \lambda\hat{\sigma}$, two concerns follow.

(1) Each weight $w_i$ ought to be non-negative for stable optimization (Ren et al., 2018). A conventional approach is to further project the weights into a $N$-dimensional simplex and adopt the derived probabilities, which is however dependent on the choice of distance measurement.

(2) As shown in Fig. 3 (a), simply minimizing variance inevitably incurs the examples with relatively smaller losses to shift towards the decision boundary. This poses a risk to the generalization, especially when $\ell$ is implemented with a maximum-margin principle (Elsayed et al., 2018).

Detailedly, Fig. 3 (a) serves as an illustrative diagram depicting examples with a shared ground truth label. Notably, well-classified instances with losses smaller than the average loss are positioned far from the decision boundary, yet also contribute significantly to variance components. Simply penalizing variance will pull all samples towards the average loss. Green arrows represent desirable upgrading direction, while purple arrows represent well-classified samples moving towards the decision boundary, which has a negative impact on generalization.

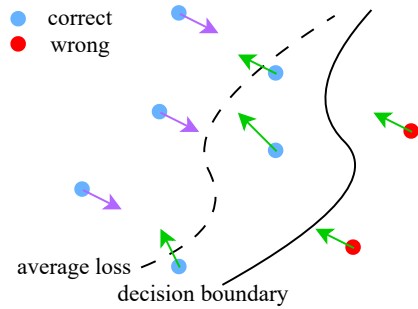

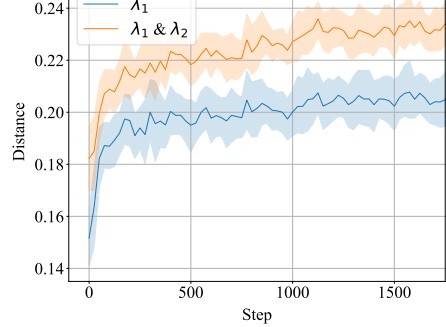

(a) How simply penalizing variance of losses can adversely affect well-classified samples.

(b) The average distance from correctly classified samples to decision boundary with and without $\lambda_2$ on the testing set across training steps.

Figure 3: The necessity and effectiveness of $\lambda_2$

To address these two concerns, we propose to directly consider the sign of $w_i$ by adjusting $\lambda$. Instead of exhaustively making every $w_i$ non-negative, for each iteration, we filter out the examples that have been confidently classified and let the weights for the remaining ones be non-negative. Specifically, if the predicted probability of the ground truth class surpasses $\tau$, which is empirically set to 0.9, the sample will be filtered out from updating in this iteration.

Let us say there remain $M$ samples that have not been confidently classified at the current iteration, and we have

$$\forall i \in [M] \quad \lambda + \frac{1}{\hat{\sigma}}(\ell_i - \hat{\mu}) \geqslant 0$$

$$\Longrightarrow \lambda \geqslant \max_{i \in [M]} \frac{\hat{\mu} - \ell_i}{\hat{\sigma}} = \frac{1}{\hat{\sigma}}\left(\hat{\mu} - \min_{i \in [M]}\ell_i\right) := \lambda_2. \tag{6}$$

Interestingly, Eq. 6 can be viewed as a conditional update to $\lambda$ based on the loss of the primary objective $\hat{\mu}$. Specifically, if $\hat{\mu}$ is close to the loss of well-classified samples, the primary objective is satisfactory and the secondary objective should be focused on. Thus $\lambda$ should be close to 0. By contrast, if $\hat{\mu}$ is relatively larger, we have a positive $\lambda$ to update the primary objective. As shown in Fig. 3 (b), the inclusion of $\lambda_2$ effectively prevents correctly classified samples from approaching the decision boundary, in contrast to those not utilizing $\lambda_2$.

By combining Eq. 4 and Eq. 6, we derive a dynamic constraint parameter that ensures harmless fairness update:

$$\lambda = \max(\lambda_1, \lambda_2, 0) = \max\left(\epsilon - \frac{\nabla\hat{\mu} \cdot \nabla\hat{\sigma}}{||\nabla\hat{\mu}||^2}, \frac{1}{\hat{\sigma}}(\hat{\mu} - \min_{i \in [M]} \ell_i), 0\right). \tag{7}$$

### 3.4 Implementation

Regarding the implementation of Eq. 7, it necessitates the computation of gradients for both $\hat{\mu}$ and $\hat{\sigma}$, as well as determining the mean loss (i.e., the value of $\hat{\mu}$) and the minimum loss value within the training samples which have not been confidently classified. Nonetheless, computing gradients across the entire dataset is time-intensive and fails to leverage the benefits offered by mini-batch processing. During the implementation phase, we compute the gradients and choose the minimum loss within the mini-batch. However, when it comes to the mean loss for variance computation, it's worth noting that the mean loss encompasses global information that could guide the update direction for each sample. As such, it should not be calculated based solely on the mini-batch. Consequently, we employ Exponential Moving Average (EMA) as an approximation for the global mean loss:

$$\hat{\mu}^t = \beta\hat{\mu}^{t-1} + \frac{1-\beta}{b}\sum_{i=1}^{b}\ell_i, \tag{8}$$

where the decay parameter $\beta$ is set 0.99 for all datasets, and $b$ denotes batch size. The detailed algorithm is left to Appendix D.

## 4 Experiments

### 4.1 Experimental Setup

**Datasets.** Four benchmark datasets encompassing both binary and multi-class classification tasks are employed. (i) UCI Adult (binary classification) (Asuncion & Newman, 2007): Predicting whether an individual's annual income is above or below 50,000 USD. (ii) Law School (binary classification) (Wightman, 1998): Predicting the success of bar exam candidates. (iii) COMPAS (binary classification) (Barenstein, 2019): Predicting recidivism for each convicted individual. (iv) CelebA (multi-classification) (Liu et al., 2015): By combining certain binary attributes, we design different multi-classification tasks, such as the prediction of hair color and beard type. Following the convention of Lahoti et al. (2020), we select sex (or gender) and race (or Young on CelebA) as sensitive attributes, splitting each dataset into 4 groups.

**Metrics.** During the evaluation phase, we gain access to the sensitive attributes that partition the dataset into $K$ disjoint groups. As discussed in Section 3.1, our training objective is to uphold a high level of overall predictive accuracy while simultaneously minimizing group accuracy disparities to the greatest extent feasible. Henceforth, we assess the performance of our method across five distinct metrics: (i)**ACC**: The overall prediction accuracy. (ii) **WACC**: The minimum group accuracy among all $K$ groups, i.e., WACC $= \min_{k \in [K]} a_k$. (iii) **MAD**: Maximum accuracy disparity, as described in Section 3.1. (iv) **TAD**: Total accuracy disparity. TAD $= \sum_{k \in [K]}(a_k - \bar{a})$, where $\bar{a}$ is the average group accuracy. (v) **VAR**: The variance of prediction loss. Since we are not able to exhaustively enumerate all possible sensitive attributes and test fairness via the metrics (ii-iv), VAR necessarily serves as a fairness proxy for any other possible selected sensitive attributes during the test phase.

**Baselines.** We simply represent our method as VFair, and compare it against four baselines including ERM, DRO (Hashimoto et al., 2018), ARL (Lahoti et al., 2020), and FairRF (Zhao et al., 2022). Note that DRO and FairRF are slightly different from our group fairness setting. DRO necessitates the identification of the worst group through a bound of group ratio, which is viewed as side information

about the protected group. FairRF selects some observed features as pseudo-sensitive attributes, constraining its application to image datasets. The detailed model structure is left in Appendix E.

## 4.2 PERFORMANCE COMPARISON

To ensure the reliability of the findings, we repeated all the experiments 10 times and averaged the outcomes. The summarized results are presented in Table 1. The best result is highlighted in red, while the second-best result is in blue. The standard deviation is shown in the bracket as $\sigma$ except for CelebA, whose $\sigma$ is smaller than 1e-4. Further comparison experiments are left in Appendix F.

Table 1: Comparison of overall accuracy and group fairness on four benchmark datasets.

|  |  | ACC $\uparrow_{(\sigma)}$ | WACC $\uparrow_{(\sigma)}$ | MAD $\downarrow_{(\sigma)}$ | TAD $\downarrow_{(\sigma)}$ | VAR $\downarrow_{(\sigma)}$ |
|---|---|---|---|---|---|---|
| UCI Adult | ERM | 84.67%(0.58%) | 80.20%(0.82%) | 16.13%(0.82%) | 20.78%(0.99%) | 0.3389(4.77%) |
|  | DRO | 84.71%(0.26%) | 80.34%(0.36%) | 15.76%(0.35%) | 20.92%(0.24%) | 0.1900(2.05%) |
|  | ARL | 84.60%(0.63%) | 80.11%(0.91%) | 16.17%(1.05%) | 20.91%(0.95%) | 0.3618(8.41%) |
|  | FairRF | 84.27%(0.13%) | 80.01%(0.15%) | 15.73%(0.18%) | 20.26%(0.58%) | 0.2583(1.38%) |
|  | VFair | 84.74%(0.34%) | 80.36%(0.49%) | 15.71%(0.73%) | 20.71%(0.80%) | 0.0817(0.98%) |
| Law School | ERM | 85.59%(0.67%) | 74.49%(1.84%) | 12.08%(2.74%) | 21.50%(3.35%) | 0.3695(1.37%) |
|  | DRO | 85.37%(0.88%) | 74.76%(2.08%) | 11.53%(1.91%) | 20.83%(2.37%) | 0.2747(1.43%) |
|  | ARL | 85.27%(0.71%) | 74.78%(2.12%) | 11.52%(2.21%) | 21.52%(1.97%) | 0.3795(1.80%) |
|  | FairRF | 81.91%(0.27%) | 68.75%(1.61%) | 14.48%(1.65%) | 26.84%(2.20%) | 0.3080(1.59%) |
|  | VFair | 85.40%(0.99%) | 75.25%(1.51%) | 11.00%(1.92%) | 19.91%(2.43%) | 0.0629(0.24%) |
| COMPAS | ERM | 66.70%(0.66%) | 63.20%(1.64%) | 07.15%(1.46%) | 09.12%(1.79%) | 0.1563(3.38%) |
|  | DRO | 66.37%(0.50%) | 62.41%(1.27%) | 07.51%(1.08%) | 09.58%(1.77%) | 0.1535(2.39%) |
|  | ARL | 66.65%(0.55%) | 63.27%(1.99%) | 06.93%(1.83%) | 09.09%(3.71%) | 0.1442(3.64%) |
|  | FairRF | 62.90%(0.43%) | 61.55%(1.06%) | 02.64%(1.55%) | 03.69%(2.10%) | 0.0693(1.26%) |
|  | VFair | 66.80%(0.27%) | 63.86%(0.57%) | 06.25%(0.80%) | 08.47%(1.23%) | 0.0186(0.12%) |
| CelebA | ERM | 92.80% | 89.77% | 03.64% | 04.77% | 0.4008 |
|  | DRO | 93.04% | 90.16% | 03.27% | 04.74% | 0.3190 |
|  | ARL | 93.26% | 89.84% | 04.02% | 05.41% | 0.3738 |
|  | FairRF | - | - | - | - | - |
|  | VFair | 93.43% | 91.09% | 02.74% | 03.85% | 0.1170 |

**Achieving small group accuracy disparity.** Our method VFair consistently outperforms other baselines on MAD and TAD except for three instances, where FairRF achieves the best. But please note FairRF sacrifices ACC significantly, showing a remarkable performance drop of around $4\%$ on COMPAS and Law School compared to others, which is not desirable under the harmless fairness setting. We also observe that our method reduces more accuracy disparities on the CelebA dataset. A reasonable explanation is that our method has the chance to find better solutions in a relatively larger solution space, where more diverse minimums can be examined by fairness criteria.

**Maintaining high accuracy.** Our method VFair also achieves competitive and even better ACC and WACC compared to baselines, representing the success in improving worst-case group performance while maintaining high overall accuracy. This highlights the effectiveness of the proposed harmless update approach. In particular, regarding the case of the COMPAS dataset whose labels are found to be noisy (Lahoti et al., 2020), Max-Min fairness methods, i.e., DRO, is prone to amplify such noises and eventually achieve worse performance than ERM. ARL remedies this issue by relying on an extra model to help identify unreliable examples. Our dynamic parameter $\lambda$ can prevent the model from excessively concentrating on noisy examples that produce large losses, which is further discussed in the ablation study (See Section 4.3). On the CelebA dataset, which is a more complex image dataset, our model achieves the highest accuracy without deepening the model layers. In contrast, ARL, which has a larger model capacity due to the addition of an adversarial network, takes second place, demonstrating the effectiveness of our method.

**Efficiency in decreasing the variance of losses.** Our method VFair showcases a substantial reduction in the variance of losses, proving that: 1) penalizing the variance of losses on the training set can effectively generalize to the testing set (Similar to Fig. 1 (a), we depict the training losses for

the rest of three datasets in Fig. 4). 2) other partitions through randomly selected sensitive attributes also enjoy better fairness results. From Fig. 4, we surprisingly see that across different datasets our VFair is the unique one that has the flattened curve while DRO, ARL, and FairRF are essentially close to ERM.

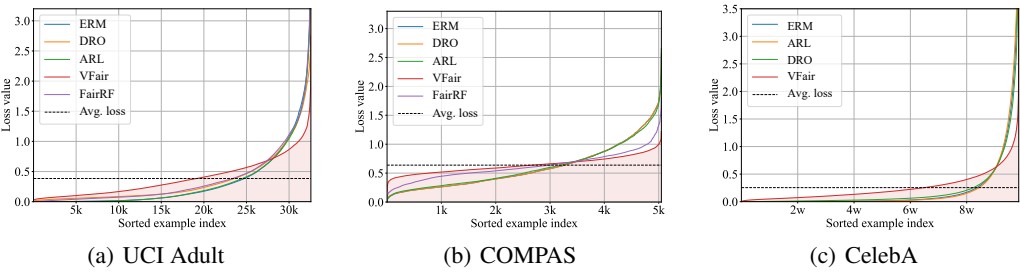

(a) UCI Adult       (b) COMPAS       (c) CelebA

Figure 4: Per-example losses for all compared methods sorted in ascending order on the training set of UCI Adult, COMPAS, and CelebA, respectively. Dash lines represent their average losses.

## 4.3 A CLOSER LOOK AT VFAIR

**Method efficacy.** We monitor the performance of VFair every 5 steps during the training phase by evaluating it with four accuracy-related metrics on the test set. The results are shown in Fig. 5, which indicate these curves naturally improve in the desired direction under the variance penalty, thus verifying the effectiveness of our method. One can also move to Appendix G for the experiments on CelebA where the training loss and test performance can be compared directly.

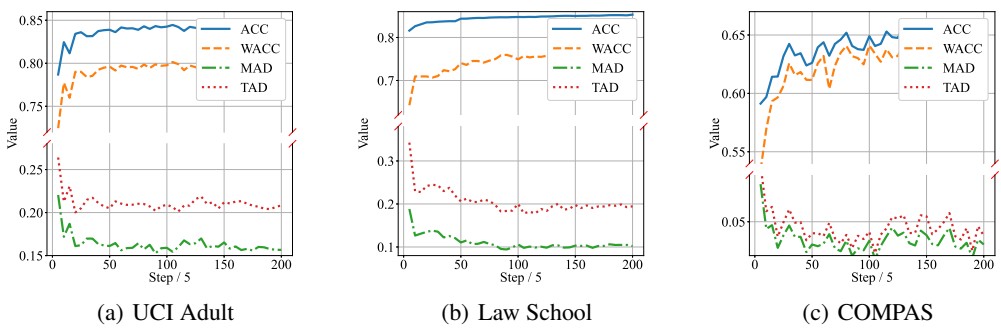

(a) UCI Adult       (b) Law School       (c) COMPAS

Figure 5: Test performance curves of four accuracy-related metrics during the training process on three benchmark datasets. The results on CelebA are left to Appendix G

**Ablation study.** We train our model under four settings: $\lambda = 1$, $\lambda = \max(\lambda_1, 0)$, $\lambda = \max(\lambda_2, 0)$, and $\lambda = \max(\lambda_1, \lambda_2, 0)$. As presented in Table 2, our method already achieves competitive results by solely employing $\lambda_2$. The full version which integrates both $\lambda_1$ and $\lambda_2$ demonstrates more stable results. Notably, on the Law School and COMPAS datasets, there exist situations when the model converges towards a uniform classifier, as indicated by the gray region. These uniform classifiers predict all the samples near the decision boundary, causing their losses to share very similar values and form variances at a scale of around 1e-7. This phenomenon underscores the effectiveness of $\lambda_2$ in preventing the model from collapsing to a low-utility model. One can refer to Fig. 7 for a more vivid explanation. Moreover, by adding $\lambda_1$, our model consistently improved in four accuracy-related metrics. These results show that $\lambda_1$ effectively guides the model to converge to a better point at the gradient level.

**Model examination.** We further examine the learned fair models through VFair by studying their parameter similarity and prediction similarity with ERM. The results (See Appendix H) demonstrate that VFair takes a chance to explore a large model space when it is necessary.

Table 2: Ablation experiments on four benchmark datasets. All of the results are averaged over 10 repeated experiments to mitigate randomness, with the best results highlighted in red and the second-best in blue (excluding the uniform situation).

| | $\lambda_1$ | $\lambda_2$ | ACC $\uparrow_{(\sigma)}$ | WACC $\uparrow_{(\sigma)}$ | MAD $\downarrow_{(\sigma)}$ | TAD $\downarrow_{(\sigma)}$ | VAR $\downarrow_{(\sigma)}$ |
|---|---|---|---|---|---|---|---|
| | $\lambda=1$ | | 84.71%(0.32%) | 80.36%(0.44%) | 15.85%(0.65%) | 20.94%(0.72%) | 0.0300(0.32%) |
| UCI Adult | | ✓ | 84.68%(0.36%) | 80.27%(0.50%) | 15.91%(0.63%) | 20.97%(0.89%) | 0.0775(0.81%) |
| | ✓ | | 84.52%(0.44%) | 80.08%(0.65%) | 15.99%(0.73%) | 20.92%(0.56%) | 0.0658(1.15%) |
| | ✓ | ✓ | 84.74%(0.34%) | 80.36%(0.49%) | 15.71%(0.73%) | 20.71%(0.80%) | 0.0817(0.98%) |
| | $\lambda=1$ | | 84.36%(0.11%) | 74.30%(0.84%) | 10.88%(0.95%) | 20.73%(1.63%) | 0.0005(0.02%) |
| Law School | | ✓ | 85.40%(0.30%) | 75.09%(0.58%) | 11.20%(0.82%) | 20.43%(1.66%) | 0.0630(0.14%) |
| | ✓ | | 45.39%(28.53%) | 32.03%(14.79%) | 30.31%(3.41%) | 53.12%(5.08%) | 0(0%) |
| | ✓ | ✓ | 85.40%(0.99%) | 75.25%(1.51%) | 11.00%(1.92%) | 19.91%(2.43%) | 0.0629(0.24%) |
| | $\lambda=1$ | | 55.21%(2.43%) | 49.90%(3.44%) | 10.51%(4.34%) | 13.28%(5.51%) | 0(0%) |
| COMPAS | | ✓ | 64.29%(0.99%) | 60.44%(3.63%) | 7.34%(3.76%) | 9.67%(4.87%) | 0.0003(0.02%) |
| | ✓ | | 66.45%(0.85%) | 63.49%(1.90%) | 6.60%(2.40%) | 8.40%(3.12%) | 0.0191(0.24%) |
| | ✓ | ✓ | 66.80%(0.27%) | 63.86%(0.57%) | 6.25%(0.80%) | 8.47%(1.23%) | 0.0186(0.12%) |
| | $\lambda=1$ | | 92.04% | 89.22% | 3.66% | 4.65% | 0.1269 |
| CelebA | | ✓ | 93.46% | 90.62% | 3.49% | 4.67% | 0.1161 |
| | ✓ | | 93.23% | 90.08% | 3.70% | 5.14% | 0.0753 |
| | ✓ | ✓ | 93.43% | 91.09% | 2.73% | 3.85% | 0.1170 |

## 5 DISCUSSION

Note that the objective of Eq. 1 that minimizes the empirical risk as well as the variance of losses looks similar to variance-bias research (Maurer & Pontil, 2009). Following Bennett's inequality, the expected risk can be upper bounded by the empirical risk plus a variance-related term (also a small variable) with a high probability:

$$\mathbb{E}_z[\ell(z;\theta)] \leqslant \frac{1}{N}\sum_{i=1}^{N}\ell(z;\theta) + C_1\sqrt{\frac{\mathbb{V}_z[\ell(z;\theta)]}{N}} + \frac{C_2}{N}, \tag{9}$$

where $C_1$ and $C_2$ are some constants which reflect the confidence guarantee. We emphasize Eq. 9 is apparently distinct from our fairness motivation by recalling Eqs. 1 and 2. Interestingly, a convex surrogate of the right-hand side of Eq. 9 is presented in Namkoong & Duchi (2017), serves as the main methodology of DRO (Hashimoto et al., 2018), providing supportive evidence of our claim: **small variance, big fairness**.

Recent studies (Hashimoto et al., 2018; Lahoti et al., 2020; Martinez et al., 2021; Chai & Wang, 2022a) that follow Rawlsian Max-Min fairness without demographic all can be summarized to essentially upweight the training examples whose losses are relatively larger. We have pointed out through Eq. 5 that our minimizing the variance of losses arrives at the same place, indicating that variance penalty also achieves a form of *implicit* examples re-weighting scheme (Lin et al., 2022). This observation further underscores the significance of our dynamic parameter $\lambda$, which mitigates the drawbacks of plainly penalizing variance.

## 6 CONCLUSION

In summary, we have introduced a straightforward yet effective variance-based method VFair that accomplishes harmless fairness without demographic information. Our approach harnesses the principle of decreasing the variance of losses to steer the model's learning trajectory, thereby bridging the utility gaps appearing at potential group partitions. The optimization with a devised dynamic weight parameter operated at both the loss and gradient levels, ensuring the model converges at the fairest point within the optimal solution set. The comprehensive experimental analysis underscores the efficacy of our method, which succeeds in attaining elevated fairness metrics while upholding model utility. Future work may explore if VFair works on other tasks, e.g., regression.

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

## A  INSTANCE LOSSES FOR REDUCING GROUP ACCURACY DISPARITY

Note that MAD $= 0$ can be also understood as encouraging the group accuracy $a$ and group index $k$ to be independent conditional on the sensitive attribute $s$, i.e., $a \perp k | s$. Without presuming what a sensitive attribute is interested in testing, we show in the following proposition the desired fairness can be instance-wise characterized.

**Proposition 2** *For any $s$ that splits data into $K$ groups, $a \perp k | s$ holds if and only if the loss $\ell$ is (approximately) independent of the training example $z$, i.e., $\ell \perp z$.*

*Proof.* We prove the statement by the following two steps.

Step 1. Since proving "$\forall s, a \perp k | s \Rightarrow \ell \perp z$" is difficult, we consider its contrapositive, i.e., "$\ell \not\perp z \Rightarrow \exists s, a \not\perp k$". If the value of $\ell$ spreads across a large range, indicating some examples are correctly (small losses) classified while others are not (large losses), we can simply let $s$ split them according to if correctly classified. Since $a_1 \neq a_2$, $a \not\perp k$ follows.

Step 2. The assertion, "$\ell \perp z \Rightarrow \forall s, a \perp k$", is correct when the condition $\ell \perp z$ is strictly satisfied, as exemplified by the scenario of a uniform classifier. When $\ell$ exhibits approximate dependence on $z$, two distinct scenarios arise. (i) All losses are concentrated in proximity to the decision boundary, resembling the characteristics of a uniform classifier. In the context of a finite partition by $s$, the accuracy of each subgroup within a uniform classifier statistically converges towards 0.5. (ii) All losses are conspicuously distanced from the decision boundary, akin to an ideal classifier. In this case, an ideal classifier consistently achieves a subgroup accuracy of 1, irrespective of the chosen split. In both situations, we can indeed conclude that $\forall s, a \perp k | s$. $\qquad\square$

Thereby, we bypass the unobserved sensitive attribute $s$ in the model training phase by requiring the loss value to be approximately independent of its input, e.g., $\ell$ is restricted in a small range.

## B  MORE ABOUT PROPOSITION 1

### B.1  PROOF OF PROPOSITION 1

*Proof.* Let $r_k$ denote the expected loss of $k$-th group. We have

$$\ell_{\text{MAD}} := \max_{k \in [K]} r_k - \min_{k \in [K]} r_k \leqslant \max_{i \in [N]} \ell_i - \min_{i \in [N]} \ell_i$$

$$\leqslant \sum_{i=1}^{N-1} |\ell_i - \ell_{i+1}| \tag{10}$$

$$\leqslant \sum_{i<j}^{N} |\ell_i - \ell_j| \overset{①}{\leqslant} \sqrt{C_N^2 \sum_{i<j}^{N} |\ell_i - \ell_j|^2}$$

$$\overset{②}{=} N \sqrt{C_N^2 \mathbb{V}_z[\ell(z;\theta)]} \tag{11}$$

$$\overset{③}{\leqslant} \frac{C_N^2 N^2}{L} \mathbb{V}_z[\ell(z;\theta)] \tag{12}$$

where ① follows the norm inequality of $||x||_1 \leqslant \sqrt{dim(x)}||x||_2$, ② uses the fact that $\mathbb{V}_z[\ell(z;\theta)] = \frac{1}{N^2} \sum_{i<j}^{N} (\ell_i - \ell_j)^2$, and ③ is similar to ① and further uses $\sum_{i<j}^{N} |\ell_i - \ell_j| \geqslant L$. $\qquad\square$

### B.2  OPTION OF LOSS FOR FAIRNESS OBJECTIVE

According to Proof B.1, Eqs. (10), (11), and (12) all upper bound the maximum group loss disparity $\ell_{\text{MAD}}$. We denote these objective as $\pi$, $\hat{\sigma}^2$, and $\hat{\sigma}$ respectively:

- $\pi = \sum_{i=1}^{N-1} |\ell_i - \ell_{i+1}|$

- $\hat{\sigma} = \frac{1}{\sqrt{N}} \sqrt{\sum_{i=1}^{N} (\ell_i - \hat{\mu})^2}$

- $\hat{\sigma}^2 = \frac{1}{N} \sum_{i=1}^{N} (\ell_i - \hat{\mu})^2$

Here, we analyze the choice of the objective from both theoretical and experimental levels. The experiment results are shown in Table 3.

Table 3: Comparison of objective selection on five fairness metrics and four benchmark datasets.

|  | objective | ACC ↑ | WACC↑ | MAD↓ | TAD↓ | VAR↓ |
|---|---|---|---|---|---|---|
| UCI Adult | $\pi$ | 82.98% | 78.35% | 16.19% | 21.10% | **0** |
|  | $\hat{\sigma}^2$ | 84.70% | 80.34% | 15.72% | 20.79% | 0.0718 |
|  | $\hat{\sigma}$ | **84.74%** | **80.36%** | **15.71%** | **20.71%** | 0.0817 |
| Law School | $\pi$ | 84.05% | 72.96% | 11.92% | 22.51% | **0.0003** |
|  | $\hat{\sigma}^2$ | 85.33% | 74.60% | 11.67% | 20.91% | 0.0691 |
|  | $\hat{\sigma}$ | **85.40%** | **74.81%** | **11.24%** | **20.31%** | 0.1935 |
| COMPAS | $\pi$ | 55.78% | 51.60% | 8.70% | 12.24% | **0** |
|  | $\hat{\sigma}^2$ | 63.45% | 59.14% | 8.71% | 11.36% | 0.0004 |
|  | $\hat{\sigma}$ | **66.80%** | **63.86%** | **6.25%** | **8.47%** | 0.0186 |
| CelebA | $\pi$ | 44.88% | 20.16% | 49.16% | 53.85% | **0** |
|  | $\hat{\sigma}^2$ | 92.45% | 89.53% | 3.44% | 4.63% | 0.1440 |
|  | $\hat{\sigma}$ | **93.43%** | **91.09%** | **2.73%** | **3.85%** | 0.117 |

**Pairwise difference objective.** We denote Eq. 10 as $\pi = \sum_{i=1}^{N-1} |\ell_i - \ell_{i+1}|$ for pairwise difference objective.

*Derivation of $\lambda_2$.* Similar to Eq. 6, the $\lambda_2$ is set to guarantee samples won't be assigned negative weights.

$$\nabla \pi = \nabla \sum_{i=1}^{N-1} |\ell_i - \ell_{i+1}| = \sum_{j=1}^{N} \sum_{i=1}^{N-1} |\ell_i - \ell_{i+1}| \frac{\partial \ell_j}{\partial \theta} = \sum_{j=1}^{N} \phi_j \frac{\partial \ell_j}{\partial \theta} \quad (13)$$

By discussing the possible combination of the absolute equation, it's obvious that $\phi_1, \phi_N \in \{-1, 1\}$, and $\phi_i \in \{-2, 0, 2\}, \forall i \in (1, N)$. Consequently, $\lambda_2$ can be simply calculated as:

$$\forall i \in [N] \quad \lambda + \phi_i \geq 0 \implies \lambda \geq 2 := \lambda_2 \quad (14)$$

*Discussion of results.* When operated on a mini-batch, unlike $\hat{\sigma}$ and $\hat{\sigma}^2$, the pairwise difference objective does not take global information into consideration, losing the relative relationships when attempting to identify the worst-case group (also discussed in Section 3.4). The instability arising from the difference of pairwise sample losses might mislead the upgrading process, as evidenced in our experiments. On challenging datasets such as COMPAS and CelebA, the model tends to converge towards a uniform classifier, even constrained by dynamic parameters.

**Deviation objective.** For Eq. 12, regarding the constants, we employ $\hat{\sigma}^2 = \frac{1}{N} \sum_{i=1}^{N} (\ell_i - \hat{\mu})^2$ as secondary regularization.

*Derivation of $\lambda_2$.* Eq. 5 change into:

$$\begin{aligned}
\nabla = \lambda \nabla \hat{\mu} + \nabla \hat{\sigma}^2 &= \frac{\lambda}{N} \sum_{i=1}^{N} \frac{\partial \ell_i}{\partial \theta} + \frac{2}{N} \sum_{i=1}^{N} (\ell_i - \hat{\mu}) \frac{\partial (\ell_i - \hat{\mu})}{\partial \theta} \\
&= \frac{\lambda}{N} \sum_{i=1}^{N} \frac{\partial \ell_i}{\partial \theta} + \frac{2}{N} \sum_{i=1}^{N} ((\ell_i - \hat{\mu})) \cdot \left( \frac{\partial \ell_i}{\partial \theta} - \frac{1}{N} \sum_{j=1}^{N} \frac{\partial \ell_j}{\partial \theta} \right) \\
&= \frac{1}{N} \sum_{i=1}^{N} (\lambda + 2(\ell_i - \hat{\mu})) \frac{\partial \ell_i}{\partial \theta} + \frac{1}{N} \underbrace{\sum_{i=1}^{N} (\ell_i - \hat{\mu})}_{=0} \cdot \sum_{j=1}^{N} \frac{\partial \ell_j}{\partial \theta}
\end{aligned} \quad (15)$$

Consequently, Eq. 6 turns into $\lambda_2 = 2\left(\hat{\mu} - \min_{i\in[K]} \ell_i\right)$

*Discussion of results.* It can be observed in ③ in Eq. 12 that $\hat{\sigma}^2$ serves as a broader constraint for $\ell_{MAD}$. As a result, it's a less restrictive objective for group disparity compared to $\hat{\sigma}$. However, the square-version term penalizes more on the variance, resulting in a more restrictive objective for the variance of losses. All of these can be proved in the experiments that using $\hat{\sigma}^2$ as objective results in lower variance but higher group disparity.

## C   Derivation of Eq. 5

Based on the form of $\hat{\mu}(\theta)$ and $\hat{\sigma}(\theta)$ in Eq. 2, we have

$$\nabla\hat{\mu} = \frac{1}{N} \sum_{i=1}^{N} \frac{\partial\ell_i}{\partial\theta} \tag{16}$$

$$
\begin{aligned}
\nabla\hat{\sigma} &= \frac{1}{2\sqrt{\frac{1}{N}\sum_{i=1}^{N}(\ell_i - \hat{\mu})^2}} \frac{1}{N} \sum_{j=1}^{N} 2(\ell_j - \hat{\mu})\frac{\partial(\ell_j - \hat{\mu})}{\partial\theta} \\
&= \frac{1}{N} \sum_{j=1}^{N} \frac{\ell_j - \hat{\mu}}{\hat{\sigma}} \frac{\partial\ell_j}{\partial\theta} - \frac{1}{N\hat{\sigma}} \sum_{j=1}^{N} \left( (\ell_j - \hat{\mu})\frac{1}{N}\sum_{k=1}^{N}\frac{\partial\ell_k}{\partial\theta} \right) \\
&= \frac{1}{N} \sum_{j=1}^{N} \frac{\ell_j - \hat{\mu}}{\hat{\sigma}} \frac{\partial\ell_j}{\partial\theta} - \frac{1}{N\hat{\sigma}} \underbrace{\left( \left(\sum_{j=1}^{N}\ell_j\right) - N\hat{\mu} \right)}_{=0} \left( \frac{1}{N}\sum_{k=1}^{N}\frac{\partial\ell_k}{\partial\theta} \right)
\end{aligned}
\tag{17}
$$

The combined gradient in Eq. 5 follows by unifiying Eq. 16 and 17.

## D   The Proposed Algorithm

The comprehensive implementation of our harmless fairness is elucidated in the following algorithm.

---

**Data:** Training set $\mathcal{D} = \{z_i\}_{i=1}^{N}$, where $z_i = (x_i, y_i) \in \mathcal{X} \times \mathcal{Y}$
**Result:** Learned model parameterized by $\theta \in \Theta$
Initialize parameters $\theta$;
Initialize $\hat{\mu}^0 \leftarrow 0$;
**for** *epoch* $\leftarrow 1$ **to** $N_{epochs}$ **do**
    **foreach** *mini-batch* $\mathcal{B} \subset \mathcal{D}$ **do**
        Compute the softmax prediction $\{s_i\}_{i=1}^{N}$;
        Filter out well-classified samples satisfy $s_i[y_i] > \tau$;
        Compute the losses for the remaining M samples $\{\ell_i\}_{i=1}^{M}$;
        Update $\hat{\mu}^t$ as in Eq. 8;
        Compute $\hat{\sigma} \leftarrow \sqrt{\frac{1}{M}\sum_{i=1}^{M}(\ell_i - \hat{\mu}^t)^2}$;
        Compute primary gradient $\nabla\hat{\mu}$;
        Compute secondary gradient $\nabla\hat{\sigma}$;
        Compute $\lambda_1$ as in Eq. 4;
        Compute $\lambda_2$ as in Eq. 6;
        Compute final gradient $\nabla$ as in Eq. 5;
        Update parameters $\theta$ as in Eq. 3;
    **end**
**end**

---

**Algorithm 1:** Harmless Fairness without Demographics via Decreasing Variance of Losses

# E  DETAILS ABOUT MODEL STRUCTURE

All the models, excluding FairRF, which operates within a distinct problem setting, conform to a shared neural network framework. Specifically, for binary classification tasks, the core neural network architecture consists of an embedding layer followed by two hidden layers, with 64 and 32 neurons, respectively. In the ARL model, an additional adversarial component is integrated, detailed in its respective paper, featuring one hidden layer with 32 neurons. For multi-classification tasks, the primary neural network transforms into resnet18, and the embedding layer transitions to a Conv2d-based frontend. Throughout these experiments, the Adagrad optimizer was employed. FairRF, utilizing its officially published code implementation, maintains the same backbone network with nuanced variations in specific details.

As For the loss function, we implemented Binary Cross-Entropy for binary classification and Cross-Entropy for multi-class classification. However, our method is general and is compatible with any other forms of loss.

# F  MORE COMPARISON EXPERIMENTS

**Maintaining superiority consistently under random partitioning.**

Given that our focus is on fairness without demographic considerations, our model excels not only in the specific divisions on attributes of race and gender but also in other possible divisions. Thus, we conducted additional experiments involving over 100 iterations, randomly splitting the data into K groups, ranking the metrics in terms of accuracy and prediction error, and calculating the average rank. For each iteration, we rank the results from the top to the bottom as 1, 2, 3, 4. It's essential to note that FairRF is not included because this method relies on selecting related attributes and is not suitable for this setting. The results shown in Table 4 indicate that our method consistently performs well regardless of the choice of K.

Table 4: Average rank of randomly splitting into 4, 10, and 20 groups.

|  |  | K=4 | | | | K=10 | | | | K=20 | | | |
|---|---|---|---|---|---|---|---|---|---|---|---|---|---|
|  |  | ACC | WACC | MAD | TAD | ACC | WACC | MAD | TAD | ACC | WACC | MAD | TAD |
| UCI Adult | ERM | 2.5 | 3.3 | 2.93 | 3.01 | 2.5 | 3.33 | 3.25 | 3.19 | 2.5 | 3.42 | 3.13 | 3.29 |
|  | DRO | 2.7 | 2 | 2.57 | 2.54 | 2.7 | 2 | 2.33 | 2.29 | 2.7 | 2 | 2.4 | 2.27 |
|  | ARL | 2.5 | 3.7 | 3.1 | 3.13 | 2.5 | 3.67 | 3.39 | 3.41 | 2.5 | 3.58 | 3.33 | 3.44 |
|  | VFair | **2.3** | **1** | **1.4** | **1.32** | **2.3** | **1** | **1.03** | **1.11** | **2.3** | **1** | **1.14** | **1** |
| Law School | ERM | **1.8** | 3.46 | 3.04 | 3.04 | **1.8** | 3.44 | 3.12 | 3.22 | **1.8** | 3.38 | 3.32 | 3.32 |
|  | DRO | 2.7 | 2.02 | 2.68 | 2.71 | 2.7 | 2.09 | 2.61 | 2.48 | 2.7 | 2.18 | 2.4 | 2.33 |
|  | ARL | 2.8 | 3.52 | 3.03 | 3.12 | 2.8 | 3.47 | 3.2 | 3.26 | 2.8 | 3.38 | 3.28 | 3.35 |
|  | VFair | 2.7 | **1** | **1.25** | **1.13** | 2.7 | **1** | **1.07** | **1.04** | 2.7 | **1.06** | **1** | **1** |
| COMPAS | ERM | 2.6 | 3.02 | 3.05 | 3.09 | 2.6 | 3.08 | 2.87 | 2.94 | 2.6 | 2.67 | 3.03 | 3.17 |
|  | DRO | 2.8 | 3.21 | 3.08 | 3 | 2.8 | 3.14 | 3.17 | 3.07 | 2.8 | 2.88 | 3.15 | 3.09 |
|  | ARL | 2.5 | 2.77 | 2.83 | 2.88 | 2.5 | 2.71 | 2.92 | 2.99 | 2.5 | 2.53 | 2.82 | 2.74 |
|  | VFair | **2.1** | **1** | **1.04** | **1.03** | **2.1** | **1.07** | **1.04** | **1** | **2.1** | **1.92** | **1** | **1** |

**Comparison with fairness methods with accessible sensitive attributes.**

We have further supplemented control experiments, where the model has access to sensitive attributes and is optimized under constrained regularization. In detail, we reproduced the MMPF in Martinez et al. (2020) and further designed experiments that penalize the losses of the minority group, denoted as PMG. Take UCI Adult and Law School as two examples, the results are presented in Table 5. By leveraging additional group information, MMPF achieves improved results. However, MMPF is not a harmless approach, particularly evident on Law School, where it sacrifices model utility for a fairer point. PMG yields unsatisfactory performance due to its excessive focus on the minority group, missing general information from other groups.

**Comparison of F1-score on CelebA.**

On CelebA, which is observed as an imbalanced multi-classification task, we further compare the F1-score of each method. Similar to the ACC-based metrics, we denote F, WF, MFD, and TFD as F1-score, Worst F1-score, Max F1-score Disparity, and Total F1-score Disparity, respectively. As

Table 5: Comparison with methods with access to attributes.

|  |  | ACC ↑ | WACC ↑ | MAD ↓ | TAD ↓ | VAR ↓ |
|---|---|---|---|---|---|---|
| | PMG | 79.48% | 73.02% | 22.36% | 27.85% | 4.1686 |
| UCI Adult | MMPF | 85.19% | 81.09% | 12.73% | 17.73% | - |
| | VFair | 84.74% | 80.36% | 15.71% | 20.71% | 0.0817 |
| | PMG | 78.52% | 70.14% | 9.47% | 15.59% | 0.8183 |
| Law School | MMPF | 82.85% | 76.20% | 8.42% | 15.43% | - |
| | VFair | 85.40% | 75.25% | 11.00% | 19.91% | 0.0629 |

shown in Table 6, our model significantly outperforms others, resulting in better utility and better fairness. In detail, the imbalanced multiclassification task setting makes the fairness problem more challenging. When evaluated under the F1-score, which considers the imbalance of each label, the worst-performing groups exhibit a larger gap with the mainstream groups, providing our method with more opportunity to address these disparities.

Table 6: Comparison of F1-score on CelebA.

|  | F↑ | WF ↑ | MFD ↓ | TFD ↓ |
|---|---|---|---|---|
| ERM | 91.40% | 70.17% | 19.39% | 22.82% |
| DRO | 91.57% | 70.16% | 20.68% | 25.25% |
| ARL | 91.60% | 70.39% | 20.14% | 24.33% |
| VFair | **91.91%** | **75.70%** | **14.39%** | **18.50%** |

## G  MORE CURVES DURING TRAINING

Fig. 6 (a) illustrates the convergence of training loss on the CelebA datasets. As the final combined objective is updated directly at the gradient level, which does not have a unified loss form, we display the curve of losses of the primary objective, representing the model's utility. Combining with 6 (b), we observe that our upgrading method in Equation 5 effectively steers the model towards convergence.

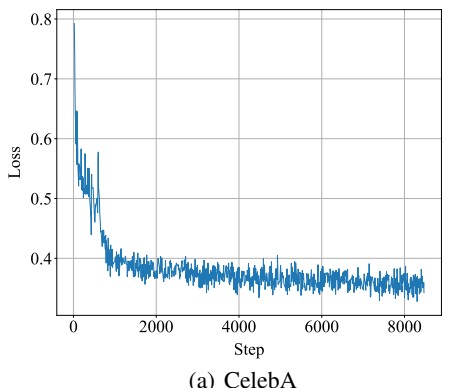

(a) CelebA

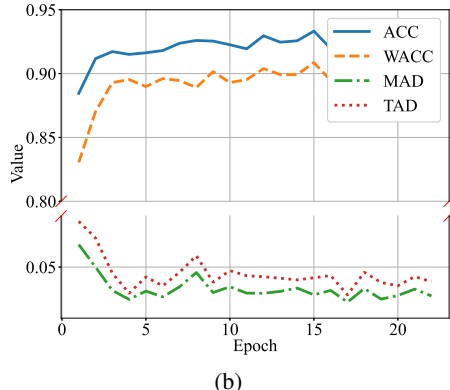

(b)

Figure 6: Test performance curves during the training process on CelebA.

To better illustrate the trend of $\lambda$ changes during training, we plotted the curves of $\lambda_1$ and $\lambda_2$ on the Law School dataset. As shown in Fig. 7, this visualization provides a vivid explanation for the grayed block in Table 2. In this special case, $\lambda_2$ is consistently higher than $\lambda_1$, preventing the model from collapsing into a uniform classifier.

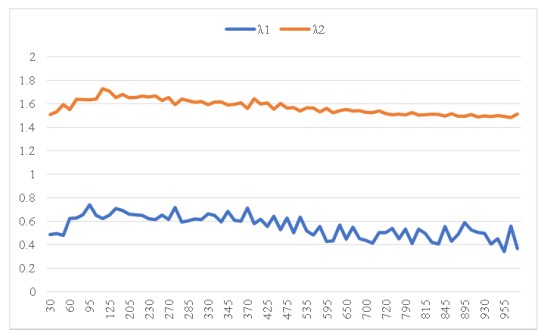

Figure 7: Curves of $\lambda$ during training on Law School

# H MODEL SIMILARITY WITH ERM.

**Harmless implementation of baselines.** To compare all baselines under the harmless fairness setting, we implement them into the same scheme and select the step with the nearest loss compared to a converged ERM. Detailedly, each method has an empirical loss, which in our method is denoted as $\hat{\mu}$ and in ARL is denoted as learner loss (compared to adversarial loss). Based on this loss, we select the harmless step which has the nearest loss value compared to a well-trained ERM model.

**Similarity with ERM.** Under the consistent harmless fairness setting, we examine the similarity between a fair model and an ERM model. We conduct experiments comparing ERM, DRO, and VFair, as they share the same model structure. By calculating the Cosine similarity and the predicted label similarity between the parameters of a fair model and ERM, we get results shown in Table 7.

Table 7: Similarity between a fair model and an ERM model.

|  | Cosine Similarity | | Prediction Similarity | |
|---|---|---|---|---|
|  | DRO | VFair | DRO | VFair |
| UCI Adult | 0.9956 | 0.9955 | 91.75% | 95.55% |
| Law School | 0.7693 | 0.5839 | 95.25% | 94.97% |

The results indicate that though these models share similar overall average losses, their model similarity varies across different tasks. This can be explained that there exist different minimums in the whole model space, and our method VFair is able to find a relatively fairer one. The prediction similarity also shows the upper bound of the value of fairness metrics under a harmless setting, like MAD.

