# OpenReview forum: "Small Variance, Big Fairness: A Path to Harmless Fairness without Demographics"
_ICLR.cc/2024/Conference — Submitted to ICLR 2024_

### Official Review · Reviewer_DML4 · 2023-10-17

**Soundness:** 3 good
**Presentation:** 3 good
**Contribution:** 3 good
**Rating:** 6
**Confidence:** 3

**Summary:**

The authors consider the problem of training a fair model with comparable accuracy on all protected groups. The catch is that the model doesn't know which observations belong to which protected groups during the training process. The main idea of the paper is to achieve low loss for each observation during training. This means minimizing the expected loss across the observations *and* the variance of the loss across the observations. The authors formulate this update as a Lagrangian and describe some heuristics for updating in a "harmless" way where the variance can decrease without impacting the expectation too much. They run experiments on four different data sets and show that their method achieves comparable accuracy to other approaches but can achieve lower variance and fairness. They also perform some ablation experiments to show their heuristic updates improve performance.

**Strengths:**

* The idea of minimizing the variance of losses is interesting.

* I like Proposition 2 that losses have to be independent of protected groups. I would like to see it in the main body and the conclusion that, since protected groups aren't known, a very natural thing to do is minimize the variance in all losses.

* The way of choosing the harmless update shows substantial thought but I would've liked more explanation.

* The experiments are comprehensive: several data sets, baselines, and metrics.

**Weaknesses:**

* I found the motivation for why they want to minimize the variance of loss weak a little weak in the main body. Moving Proposition 2 to the main body and adding more explanation could help.

* I found Proposition 1 unpersuasive because the squareroot of the variance is a *very* loose upper bound on the maximum accuracy difference loss. At one point in the proof, they blow up one side by a factor of n. I would only find a result like this persuasive if there was a (somewhat) tight upper and lower bound.

* Honestly, the results in Table 1 aren't impressive to me. Most of the metrics are quite similar except for variance which, to be fair, only their algorithm has been optimized for.

**Questions:**

Is my assessment of Proposition 1 as loose correct?

Is my assessment of the algorithms having comparable performance in the experiments correct?

Can the problem be sold in the context of federated learning? It seems like a natural fit there where you're training on a private distribution and you want your model to perform well on other distributions that you don't have access to.

Do you have results for the time it took to train each model (sorry if I missed this)? It seems like your training method uses more information which might be an unfair advantage over other methods if it takes significantly longer.

---

> ### Author Response · Authors · 2023-11-18
>
> Dear Reviewer,
>
> We appreciate your thoughtful review of our paper. Your feedback is instrumental in refining our work, and we appreciate the time and consideration you have given to our manuscript. Here are our responses to your comments and questions:
>
> 1. **The position of Proposition 2**
>
> + Regarding the motivation of minimizing the variance of losses, we suggest readers also refer to the **Idea**​ paragraph in the Introduction section, where variance is also viewed as the second moment of loss distribution. We thought it is straightforward and useful to character accuracy disparities when demographics are unknown.
> + Note that Proposition 2 is derived by using MAD metric, and it also expects that loss should be (approximately) independent of input, which is used to help understand the problem only. Thus, we left it to the appendix as a supplementary.
>
> 2. **Concern about Proposition 1**
>
> Thank you for your detailed reading.
>
> + In Appendix B, we presented that Eq. (10) formulated as a pairwise loss which is tighter than variance without blowing up one side by a factor of $n$. However, shown as in Table 3 on Page 13, the pairwise loss has not achieved better results because the global information (e.g., mean loss) is hard to be incorporated for a minibatch update in practical implementation.
> + Although loose, experimental results in Table 3 demonstrated that variance minimizing is useful to characterize accuracy disparity without demographics. We understand tight bound is a pursuit in most cases, while it is not the only criterion in this scenario.
>
> 3. **Experimental Results**
>
> We have conducted numerous detailed new experiments, please refer to the global comments for more information.
>
> 4. **Relationship with Federated Learning**
>
> We understand that learning on data without demographics is closely related to the data privacy community. However, please note that our problem setting does not introduce any prior about demographics, which indicates the solution should adapt to any demographic distribution. Thus, there is no need to set distributed groups that are based on federated learning. If each local group is exactly a group divided by interested attributes, then the central model at least knows how many local groups there exist (when it executes gradient aggregation), which can be some side information to demographics and thus beyond our problem setting.
>
> 5. **Training time of each method**
>
> Each method shares similar training steps. While explicit details of the training steps are not provided, a comprehensive account of the experimental configurations is presented in Appendix F, specifically within the section titled "Harmless Implementation of Baselines." In this section, we meticulously elucidate the selection of steps within a harmless fairness context. To expound further, each method is characterized by an empirical loss denoted as $\hat{\mu}$ in our methodology and as the learner loss (in comparison to adversarial loss) in the ARL approach. This empirical loss serves as the criterion for step selection, whereby we opt for the harmless step that manifests the loss value most proximate to that of a well-trained ERM model. This methodological choice ensures a judicious alignment with established benchmarks, fostering a fair evaluation of the experimental outcomes.
>
> We hope these planned revisions and clarifications will address your concerns. Please let us know if you have any further questions. We are inclined to address them to improve the quality of this work.

---

> > ### Comment · Reviewer_DML4 · 2023-11-22
> >
> > 1. **The position of Proposition 2**
> >
> > Totally your purview about where the proof goes. Personally, I found the proof to be elucidating and reading it was when I started to get excited about your work.
> >
> > 2. **Concern about Proposition 1**
> >
> > Hard to discern from your response but it sounds like you agree that Proposition 1 is quite loose in particular the step in Equation 10. I would *strongly* encourage you to make this clear when you state Proposition 1 in the body. Otherwise, it is quite misleading.
> >
> > 4. **Relationship with Federated Learning**
> >
> > My comment was only a suggestion about a potential application of your work that could make it even stronger.
> >
> > 5. **Training time of each method**
> >
> > From your response, it sounds like you use approximately the same number of training steps? That's not necessarily the same thing as time because maybe each one of your training steps takes longer. I would really like to see the total training time for your model in comparison to the other ones to address this concern.
> >
> > Based on your current responses, I will keep my evaluation.

---

> > > ### Author Response · Authors · 2023-11-23
> > >
> > > We thank the reviewer for taking time to read our rebuttal. Your comments are valuable to improve the quality of this paper. However, we apologize that the corresponding changes would be not instantly updated in the uploaded pdf, and we prefer revising the paper afterwards by considering all other comments.
> > >
> > > 1. **The position of Proposition 2**
> > >
> > > We appreciate your recognition about the importance of the Proposition 2. Following your suggestion, we will move Proposition 2 to the main body in the final version to make it clearer.
> > >
> > > 2. **Concern about Proposition 1**
> > >    + We will add a remark under Proposition 1 to explain that although being a bit loose, $\hat{\sigma}$ essentially takes advantage of global information (via using mean, compared to pairwise loss $\pi$) and is not as loose as the naïve variance $\hat{\sigma}^2$. We will also refer readers to the experimental justification (Table 3 in Appendix) wherein $\hat{\sigma}$ is demonstrated as the most powerful loss for achieving fairness.
> > >    + We authors have also discussed about other possible losses which could be tighter and more useful as commented by the reviewer. To the best of our knowledge, however we are not able to provide feasible solutions to this concern. We agreed this is important and would make it as a future research direction.
> > >
> > > **4.**    **Relationship with Federated Learning**
> > >
> > > Inspired by your comments, we further searched the literature and found some work [1] does connect federated learning with fairness without demographics. Based on such research, we believed that FL could be a potential application of our method.
> > >
> > > **5.**    **Training time of each method**
> > >
> > > We monitored the training time for each method and reported the results in Table 1. All of these experiments were conducted on one Nvidia RTX 3090.
> > >
> > >  **Table 1: Average training time of each method on four datasets.**
> > >
> > > |             | ERM      | DRO      | ARL      | FairRF    | VFair     |
> > > | ----------- | -------- | -------- | -------- | --------- | --------- |
> > > | UCI  Adult  | 5.469sec | 3.947sec | 8.436sec | 10.686sec | 11.238sec |
> > > | Law  School | 6.978sec | 5.695sec | 6.066sec | 7.255sec  | 7.204sec  |
> > > | COMPAS      | 1.534sec | 1.681sec | 2.069sec | 2.946sec  | 2.838sec  |
> > > | CelebA      | 46.71min | 47.44min | 49.57min | -         | 58.63min  |
> > >
> > > The results showed that training time of our method, VFair, is slightly higher than that of ERM and DRO. This is because the derivation of dynamic $\lambda$ needs to **compare** the gradients of the primary and secondary objectives before updating the model, necessitating twice the gradient calculations. ARL is time-intensive due to the introduction of an additional adversary network for computing instance weights. The training time of FairRF is comparable to ours and it uses a candidate descent approach and may not be scalable to large datasets. For a more direct comparison, all methods on CelebA were trained for 20 epochs, and we do not leverage more knowledge but use longer training times.
> > >
> > > [1] Federated Fairness without Access to Demographics, Neurips, 2022.

---

### Official Review · Reviewer_6MVk · 2023-10-30

**Soundness:** 3 good
**Presentation:** 2 fair
**Contribution:** 3 good
**Rating:** 6
**Confidence:** 3

**Summary:**

The paper studies the problem of deploying a fair machine learning model without using user demographic information. This problem emerges when demographic information is sensitive or private and, therefore, cannot be collected and stored by the model deployer. The paper proposes minimizing the variance of losses during training while roughly maintaining the average loss as a solution to improve fairness without demographics. Experiments are performed to demonstrate that this approach (variance minimization) improves fairness without demographics.

**Strengths:**

•	The paper is well written, and the related work is introduced in a very reader-friendly manner.

•	The idea of minimizing the loss variance as a surrogate for improving fairness is exciting. Specifically, it makes practitioners avoid learning group information, increasing privacy. I would appreciate it if the paper gave more intuition about why we should expect that variance minimization improves fairness.

•	This reviewer appreciates the detailed discussion and intuition about how to solve the proposed optimization problem. It was very clearly explained.

•	The results in the paper (Table 1) indicate that the simple proposed approach of decreasing variance may outperform existing methods – see weaknesses for comments.

**Weaknesses:**

•	The paper only considers accuracy disparities as fairness metrics. At the beginning of the paper, the authors mention fairness metrics such as equalized odds; however, such metrics are not considered. The authors focus on accuracy disparities and do not provide results for other fairness metrics of major interest in the literature. This reviewer strongly suggests the authors replace two of the metrics in Table 1 with other fairness metrics (e.g., equal odds and statistical parity).

•	Table 1 provides results for the performance of different fairness improvement techniques. However, the reported values are very close. Therefore, it is hard to get any conclusion out of the reported values. Can the authors please add confidence intervals?

**Questions:**

•	How does this paper relate with works in machine learning that argue that a flat loss landscape implies better generalization [1]? Does variance minimization imply a flat minimum in the loss landscape?

•	The problem of minimizing some model performance constrained to the loss being small enough has been studied in the Rashomon effect literature. The paper may benefit from related work in the field, such as [2]. Moreover, the authors argue that if $\delta$ (Eq. 1) is small enough, then the "fair" model and the ERM are equivalent. A recent paper showed how small $\delta$ needs to be to ensure that the models are equivalent [3].


[1] Hao Li et al. Visualizing the Loss Landscape of Neural Nets. 2018.

[2] Amanda Coston et al. Characterizing Fairness Over the Set of Good Models Under Selective Labels. 2021.

[3] Lucas Monteiro Paes. On the Inevitability of the Rashomon Effect. 2023.

---

> ### Author Response · Authors · 2023-11-18
>
> Dear Reviewer,
>
> We appreciate your thorough review of our paper and your insightful comments. Your feedback is invaluable in refining and strengthening our work. Here are our responses to your comments and questions:
>
> 1. **Other Fairness metrics**
>
> + As we claimed in the Introduction, utility disparity across different groups is our interest, which also aligns with the recent studies based on Rawlsian fairness where predictive accuracy of the worst-off group is concerned and they do not include equal odds and statistical parity either.
> + Note that equal odds and statistical parity measure fairness over predicted y, which requires the latent representation or final prediction to be somehow invariant to groups. On the one hand, these metrics are not compatible with the concept of model utility we advocated. On the other hand, to achieve such fairness we need to reformulate the objective because our original goal is to let loss value (determined by both predicted y and ground truth y) be approximately invariant to groups.
>
> 2. **Experimental Results**
>
> We have conducted numerous detailed new experiments, please refer to the global comments for more information.
>
> 3. **Discussion of related works**
>
> Thanks for suggesting the related works. We will incorporate relevant discussions and references in the related work section to provide a more comprehensive background for our work.
>
> + We understand variance regularization on ERM aims at improving the model reliability but may not necessarily imply a flat minimum in the loss landscape [1]. A flat minimum like sharpness-aware minimization [4] expects that average loss should not change by much if a model is perturbed within a small region. However mathematically, variance minimization cannot ensure such a property.
> + Based on the concept of Rashomon [3], research [2] also proposed a method that achieves fairness from good-utility models, which is quite similar to the branch of “Harmless Fairness” in Related Work. Our work differs from theirs by two key differences. (1) We further focused on “Harmless fairness without demographic” and thus paid much attention to how to describe fairness under such a restrictive scenario. (2) Research [2] solves a constrained optimization problem, which means we need a proper upper bound for the average loss. One may obtain it by running an ERM model beforehand and then setting it, which is not needed as we treat it as a bi-objective problem and the constraint is satisfied by adjusting the weight factor $\lambda$.
>
> We hope these planned revisions and clarifications will address your concerns. Please let us know if you have any further questions. We are inclined to address them to improve the quality of this work.
>
> [1] Hao Li et al. Visualizing the Loss Landscape of Neural Nets. 2018.
>
> [2] Amanda Coston et al. Characterizing Fairness Over the Set of Good Models Under Selective Labels. 2021.
>
> [3] Lucas Monteiro Paes. On the Inevitability of the Rashomon Effect. 2023.
>
> [4] Sharpness-Aware Minimization for Efficiently Improving Generalization, ICLR 2021.

---

> > ### Comment · Reviewer_6MVk · 2023-11-22
> >
> > Thank you, authors, for your response.
> >
> > I will maintain my score.
> > However, I want to highlight the importance of improving the paper based on the comments of reviewers 6VBP and NHS2.

---

> > > ### Author Response · Authors · 2023-11-23
> > >
> > > Thanks for taking time to read our rebuttal. We have responded to the comments of Reviewer 6VBP and NHS2, hoping our detailed explanation and additional comparison experiments can address common concerns.

---

### Official Review · Reviewer_NHS2 · 2023-10-31

**Soundness:** 2 fair
**Presentation:** 3 good
**Contribution:** 2 fair
**Rating:** 5
**Confidence:** 5

**Summary:**

To enhance fairness in situations where sensitive attributes are unavailable, this paper proposes the minimization of the variance of the loss function, specifically minimizing loss distribution’s second moment while not increasing its first moment, dubbed as VFair in this paper.

To enhance fairness without compromising the model's utility, this paper builds upon dynamic barrier gradient descent [1] and introduces further improvements. During the training process, the neural network's parameters are updated harmlessly, thereby ensuring the model's utility is preserved.

**Strengths:**

•	This paper proposes a simple yet effective method to improve fairness without demographic information, namely by minimizing the variance of the training loss.

•	Furthermore, to ensure the enhancement of fairness without compromising the model's utility, the harmless update method is used to update the model's parameters. This method ensures that gradient updates align with the primary objective's gradient without causing conflicts.

•	Point out the relationship between MAD and the variance of the loss function.

**Weaknesses:**

•	Although this is the first time introducing the idea of reducing the variance of the loss function into the fairness domain, similar concepts have been explored in earlier works [2, 3, 4]. It may be advisable to include a more comprehensive discussion in the related work section.

•	It is advisable to supplement additional experimental details. For example, a more comprehensive elaboration on the configuration of the ERM model for various datasets, particularly those not discussed in the DRO, would be beneficial. Additionally, it is better to specify the optimizer and the learning rate decay strategy, as these factors may have a impact on the stability and convergence of the loss function. While the paper briefly mentions the setting of the hyperparameter $\epsilon$, it may appear somewhat ambiguous. It is also necessary to clarify the robustness of the hyperparameter $\epsilon$.

•	The explanation of the optimization objective in Appendix B.2 seems to contradict the main body of the paper. The $\hat{\sigma}$ in the main body refers to a different object than the $\hat{\sigma}$ mentioned in the appendix.

•	While the gains over baselines are moderate to low, it is advisable to provide standard deviations to better illustrate the stability of the outcomes.

[1] Gong, Chengyue, and Xingchao Liu. "Bi-objective trade-off with dynamic barrier gradient descent." NeurIPS 2021 (2021).

[2] Balaban, Valeriu, Hoda Bidkhori, and Paul Bogdan. "Improving Robustness: When and How to Minimize or Maximize the Loss Variance." 2022 21st IEEE International Conference on Machine Learning and Applications (ICMLA). IEEE, 2022.

[3] Li, Tian, et al. "Tilted empirical risk minimization." arXiv preprint arXiv:2007.01162 (2020).

[4] Lin, Yexiong, et al. "Do We Need to Penalize Variance of Losses for Learning with Label Noise?." arXiv preprint arXiv:2201.12739 (2022).

**Questions:**

•	Is the "the sample will be filtered out from updating in this iteration." in the second-to-last paragraph on page five meaning not using this portion of the sample for training?

•	How is it derived from equation (6) that $\lambda = 0$ in certain cases?

---

> ### Author Response · Authors · 2023-11-18
>
> Dear Reviewer,
>
> We appreciate your detailed and insightful feedback on our manuscript, and we would like to express our gratitude for taking the time to thoroughly review our work. We have carefully considered each of your points and suggestions, and we are committed to addressing them to enhance the quality and clarity of our contribution.
>
> 1. **Discussion about related works**
>
> Thank you for suggesting the related studies. As we discussed in Section 5, we agreed that penalizing the variance of losses can be motivated by different applications, and we would add its advantage of robustness [1]. Particularly, in the submitted manuscript, we have pointed out the relation between variance regularization and instance-reweight learning inspired by the work [2], which is not a fairness study though. TERM [3] reformulates the standard ERM which improves the worst-performance fairness and thus would be included as a worst-case fairness method.
>
> 2. **Additional experimental details**
>
> Your suggestion regarding additional experimental details is well-received. We apologize for the omission of the model structure and will include the necessary details in the paper.
>
> + All the models, excluding FairRF, which operates within a distinct problem setting, conform to a shared neural network framework. Specifically, for binary classification tasks, the core neural network architecture consists of an embedding layer followed by two hidden layers, with 64 and 32 neurons, respectively. In the ARL model, an additional adversarial component is integrated, detailed in its respective paper, featuring one hidden layer with 32 neurons. For multi-classification tasks, the primary neural network transforms into resnet18, and the embedding layer transitions to a Conv2d-based frontend. Throughout these experiments, the Adagrad optimizer was employed. FairRF, utilizing its officially published code implementation, maintains the same backbone network with nuanced variations in specific details.
> + To address the ambiguity in the explanation of the hyperparameter epsilon, we will provide a more detailed clarification. When the angle of the gradient reaches an orthogonal angle, Formula 4 can be simplified as $\lambda \ge \epsilon$, indicating that in the absence of conflicts in gradient updating, to what extent do we still want the primary objective to be updated? **We designate this as a hyperparameter because we have observed variations in the quality of data across different datasets.** For instance, on the COMPAS dataset, known for its numerous noisy samples [4], a larger epsilon is needed to control the model to maintain utility. In this case, the regularization of the second moment of the distribution is constrained, preventing the model from focusing more on outlier samples. For reproducibility, it is recommended to set the value of epsilon as 0, 0.5, 0.5, and 3 for Law School, UCI Adult, CelebA, and COMPAS, respectively.
>
> 3. **The ambiguity of symbols**
>
> We apologize for the ambiguity of objectives. To be clearer, we used $\pi, \hat{\sigma}, \hat{\sigma}^2$ to denote the 3 objectives derived in B.1 and Table 3:
>
> + $\pi=\sum_{i=1}^{N-1}|\ell_i - \ell_{i+1}|$
> +  $\hat{\sigma} = \frac{1}{\sqrt{N}}\sqrt{\sum_{i=1}^N(\ell_i - \hat{\mu})^2}$
> + $\hat{\sigma}^2 = \frac{1}{N}\sum_{i=1}^N(\ell_i - \hat{\mu})^2$
>
> 4. **Experimental Results**
>
> We have conducted numerous detailed new experiments, please refer to the global comments for more information.
>
> 5. **Details about the filtered samples**
>
> Your interpretation is accurate. This subset of samples will not be utilized for training during the current iteration. Nevertheless, in the subsequent iteration, we will reevaluate the selection of samples
>
> 6. **The derivation of $\lambda=0$**
>
> We appreciate your inquiry and apologize for the confusion we may have caused. Actually, the statement is not derived from Equation 6. Here, we aim to provide an intuitive explanation of the update strategy. If the primary objective is satisfactory, attention should shift to the secondary objective. In this case, the lambda should be relatively small, potentially becoming zero as it is non-negative. The design of $\lambda_2$, as outlined in Equation 6, aligns with this strategy. When the primary objective is satisfactory, the value of \mu tends to be close to the minimum loss of the selected samples, resulting in a value close to zero.
>
> We hope these planned revisions and clarifications will address your concerns. Please let us know if you have any further questions. We are inclined to address them to improve the quality of this work.

---

> > ### Author Response · Authors · 2023-11-18
> >
> > [1] Balaban, Valeriu, Hoda Bidkhori, and Paul Bogdan. "Improving Robustness: When and How to Minimize or Maximize the Loss Variance." 2022 21st IEEE International Conference on Machine Learning and Applications (ICMLA). IEEE, 2022.
> >
> > [2] Lin, Yexiong, et al. "Do We Need to Penalize Variance of Losses for Learning with Label Noise?." arXiv preprint arXiv:2201.12739 (2022).
> >
> > [3] Li, Tian, et al. "Tilted empirical risk minimization." arXiv preprint arXiv:2007.01162 (2020).
> >
> > [4] Preethi Lahoti, Alex Beutel, Jilin Chen, Kang Lee, Flavien Prost, Nithum Thain, Xuezhi Wang, and Ed Chi. Fairness without demographics through adversarially reweighted learning. Advances in neural information processing systems, 33:728–740, 2020.

---

> > > ### Comment · Reviewer_NHS2 · 2023-11-23
> > > **Response to the author**
> > >
> > > Dear authors,
> > >
> > > Thanks for your reply. I agree with 6VBP and there is still some to deal with. So I stand my score.

---

> > > > ### Author Response · Authors · 2023-11-23
> > > >
> > > > Thanks for taking time to read our rebuttal. We have responded to the comments of Reviewer 6VBP, hoping our detailed explanation and additional comparison experiments can address your common concerns. We would appreciate it if you could specify any additional concerns you may have.

---

### Official Review · Reviewer_6VBP · 2023-10-31

**Soundness:** 2 fair
**Presentation:** 3 good
**Contribution:** 1 poor
**Rating:** 3
**Confidence:** 4

**Summary:**

The paper proposes minimizing the standard deviation of classification loss among training samples as a way to promote fairness without access to sensitive attributes (by promoting worst-off group accuracy).

**Strengths:**

- Ensuring worst-off group fairness for unknown group partitions has strong real-world significance.
- Using variance as a proxy for worst-off group fairness looks novel.
- The paper is well written.

**Weaknesses:**

- Experimental results look underwhelming and not accompanied by confidence intervals or any form of statistical significance tests.
  - Differences between different methods or to unconstrained ERM do not seem large; it's difficult to attribute the proper importance to the very small metric differences without this context.
  - Plus, all results correspond to averages over 10 runs, so these details should be easy to compute.

- Code or experimental artifacts are not shared.
  - Properly reviewing the paper would require access to code and empirical results, since the main claims are quite empirical in nature (e.g., lower loss variance equates to higher fairness).

- It would be important to have a comparison with methods from the constrained optimization literature (standard variance minimization subject to constraint on the empirical risk) or from the standard fairness literature (e.g., with access to sensitive attributes).
  - Of course it is not expected for VFair to surpass methods that have access to sensitive attributes, but it would be useful to properly contextualize VFair's results.

- Some important details are missing (see `Questions`).

**Questions:**

- Was a standard constrained optimization approach to the problem attempted before constructing the proposed algorithm?
  - Although the intuition behind VFair is well explained, it seems that to motivate a new algorithm it would make sense to compare to standard CO algorithms or justify in some other way why they are not used.
  - Why is the standard Lagrangian dual ascent formulation not used? e.g., see [A], and the [TensorFlow CO framework](https://github.com/google-research/tensorflow_constrained_optimization/blob/master/README.md)


- > "Lagrangian function, which however employs a fixed and finite multiplier and thus may not fully satisfy the constraint"
  - The Lagrangian function does not employ a fixed multiplier, the $\lambda$ would have to be optimized jointly with the $\theta$ until a stable saddle point is reached.

- What ML algorithm is actually used to fit the data? The paper is written w.r.t. generic model parameters $\theta$, which is fine, but what actual model was used in the experiments? A neural network?
- Possibly I missed it, but I can't seem to find an actual definition for the loss function $\ell$ used throughout the paper. Is it BCE?
  - Multiple plots show the distribution of sample-wise loss for different models; are all algorithms evaluated on the same loss function?

- Is there any explanation for why some (greyed out) rows of Table 2 always result in a constant/uniform classifier over the 10 runs?
  - Perhaps plotting $\lambda_1$ and $\lambda_2$ as well as loss as training progresses could shed some light.

---
[A]: Cotter, Andrew, Heinrich Jiang, and Karthik Sridharan. "Two-player games for efficient non-convex constrained optimization." Algorithmic Learning Theory. PMLR, 2019.

---

> ### Author Response · Authors · 2023-11-18
>
> Dear Reviewer,
>
> We would like to express our gratitude for taking the time to thoroughly review our work. We have carefully considered each of your comments, and we are committed to addressing them to enhance the overall quality.
>
> 1. **Experimental Results**
>
> We have conducted numerous detailed new experiments, please refer to the global comments for more information.
>
> 2. **Code and Experimental Artifacts**
>
> Following your suggestion, we have included our code in the supplementary materials, which showcases a simple yet effective approach to achieving fairness. We will release the full code once this paper is accepted.
>
> 3. **Comparison with Constrained Optimization Approaches**
>
> + We reminded that utilizing constrained optimization essentially requires a proper $\delta$ (an upper bound of the minimum of loss in Eq. 1&2) which is not accessible unless minimizing the average losses on the entire training set beforehand.
> + We clarified that the targeted problem is treated as a bi-objective problem in this paper, with two objectives having different priorities. In this sense, the proposed algorithm does not have to handle $\delta$ separately, while “the constraint” is implicitly satisfied. While the suggested [1] and TFCO focus on solving constrained optimization problems. To avoid confusion, we will cite these works and offer a reminder in the main body of the paper.
> + Using a fixed λ, which is simply set to 1 in Table 2, we linearly combined two objectives, which serve as an ablation study in our experiments. The results showed that the model performance may significantly drop, e.g., on COMPAS.
> + We apologize for the description of "a fixed multiplier". Our intended meaning is that the advanced Lagrange method aims at the optimal multiplier while it is dynamic along with the optimization process in our method.
>
> 4. **Details on the model structure**
>
> Following your advice, we have included the necessary description in the paper.
>
> All the models, excluding FairRF, which operates within a distinct problem setting, conform to a shared neural network framework. Specifically, for binary classification tasks, the core neural network architecture consists of an embedding layer followed by two hidden layers, with 64 and 32 neurons, respectively. In the ARL model, an additional adversarial component is integrated, detailed in its respective paper, featuring one hidden layer with 32 neurons. For multi-classification tasks, the primary neural network transforms into resnet18, and the embedding layer transitions to a Conv2d-based frontend. Throughout these experiments, the Adagrad optimizer was employed. FairRF, utilizing its officially published code implementation, maintains the same backbone network with nuanced variations in specific details.
>
> 5. **Details about Loss Function**
>
> Thank you once again for your detailed reading. Throughout the experiments, we implemented binary cross-entropy (BCE) for binary classification and categorical cross-entropy (CE) for multi-class classification. However, we would like to underscore that our method is general and is compatible with other forms of loss as well. We will incorporate this information into our main paper.
>
> 6. **Grayed Out Rows in Table 2**
>
> Table 2 shows that applying a fixed $\lambda$ or using $\lambda_1$ only might result in uniform classifiers. This is because without adjustment of $\lambda_2$, the model is at risk of providing very uncertain predictions to training samples, that is, most of the data lies near the decision boundary. Fig. 3 observes the same phenomenon by monitoring the distance between samples to the decision boundary, which demonstrates the necessity of $\lambda_2$. Mathematically, $\lambda_2$ lifts up the importance of the average loss from the “loss perspective” even if the gradients of two objectives do not conflict with each other by much. Following your advice, we added the plot of $\lambda$ in Appendix G.
>
> We hope these planned revisions and clarifications will address your concerns. Please let us know if you have any further questions. We are inclined to address them to improve the quality of this work.
>
> [1]: Cotter, Andrew, Heinrich Jiang, and Karthik Sridharan. "Two-player games for efficient non-convex constrained optimization." Algorithmic Learning Theory. PMLR, 2019.
>
> [2] Gong, Chengyue, and Xingchao Liu. "Bi-objective trade-off with dynamic barrier gradient descent." NeurIPS 2021 (2021).

---

> > ### Comment · Reviewer_6VBP · 2023-11-22
> >
> > Thank you for the rebuttal clarifications and the paper additions. I think the added details have definitely improved readability of the manuscript.
> >
> > Regarding the added standard deviation details for Table 1 (which are still missing for Table 2): are the standard deviation results in percentage (as are the values shown), or are the results in absolute terms? I find it hard to believe that a standard deviation of 0.007% can be achieved over 10 runs of a stochastic method on such a small dataset as COMPAS. If these values are not in percentage, please correct to use the same unit throughout the table. Additionally, if they are in absolute terms, it means most metrics for most models are actually pair-wise indistinguishable, except for the variance for which only VFair was optimized. This fact seems to have been pointed out by several reviewers.

---

> > > ### Author Response · Authors · 2023-11-23
> > >
> > > Thanks for your prompt reply which provides us with more chance to address your further concerns.
> > >
> > > 1. As pointed out by the reviewer, the standard deviation in Table 1 should be presented in percentage to avoid confusion. We have updated it in the manuscript. To make it more consistent, we added the standard deviation of Table 2 as well. Please refer to the latest PDF version or edited global comments.
> > > 2. We regrated creating an impression for reviewers that our method is only good on the variance metric because our model has optimized it. We want to clarify this concern from the following aspects.
> > >    + We again reminded readers that the relatively smaller improvements in Table 1 should be attributed to the use of specific attributes. We never intended to exhibit some decent performances by selecting some attributes on which our method can overwhelm others.
> > >    + As the target problem of this paper is fairness without demographics, meaning that any possible attributes are interests, we believed that variance could serve as a useful metric because zero variance suggests that for any sensitive attributes, the corresponding group disparities constantly remain zero.
> > >    + We reminded that the compared methods DRO and ARL are also implicitly optimizing variance. For example, the dual objective of DRO is minimizing $\sum_i [\ell(\theta;z_i)-\eta]_+^2$ (See Eq. 6 of paper [1]), which reduces to variance when $\eta$ takes mean loss and Relu operator is dropped.
> > >    + In the earlier uploaded rebuttal, we realized that random splitting can be a more convincing way to remove the dependence on the specific attributes and thus designed and added corresponding experiments (Table 3 of rebuttal, which is also updated as Table 4 in the manuscript).
> > >
> > > We would appreciate it if such explanations could address your concern about model performance comparison. We are also delighted to answer any future questions about this paper.
> > >
> > > [1] Tatsunori Hashimoto, Megha Srivastava, Hongseok Namkoong, and Percy Liang. Fairness without demographics in repeated loss minimization. In International Conference on Machine Learning, pp. 1929–1938. PMLR, Jun 2018.

---

### Author Response · Authors · 2023-11-18

We appreciate the valuable comments from the reviewers. In response to the common issues, we have conducted numerous thorough new experiments.

+ The tables in the main body of the paper did not include statistical results due to the limited space. Following reviewer's suggestion, we have added corresponding standard deviations. As shown in Table 1 and Table 2, our method is capable of consistently maintaining stable performance.
+ We understand that the differences in terms of accuracy-based metrics among compared methods seemed limited (Note that VFair always achieves the best performance on VAR across all datasets). Such small differences can be attributed to adopting specific attributes and discontinuous of ACC (i.e., 0-1 loss). As we can observe, all fairness methods fail to earn much on fairness metrics.
+ To make it more convincing, we conducted experiments that randomly split the datasets into K groups over 100 iterations. Results in Tables 3 show that our method **excels not only in the specific divisions on attributes of race and gender but also in other possible divisions.**
+ We compared our method with fairness methods with accessible sensitive attributes in Table 4.
+ On the imbalanced multiclass dataset CelebA, we conducted a further comparison of F1-score for each method. The results in Table 5 **demonstrate our method's significant improvement.**
+ We have made simultaneous modifications to the paper. The changes can be directly viewed in the latest PDF version, highlighted in cyan. Summarized modifications are listed below:
  + We added additional discussion on related work in Section 2.
  + Further comparison experiments with more convincing results are now presented in Appendix F.
  + Details about the model structure and loss function are included in Appendix E.
  + New curves of $\lambda$ during training have been added in Appendix G for a more vivid explanation of Table 2.
  + We edited expressions that might cause confusion.

**Table 1 Comparison of overall accuracy and group fairness on three benchmark datasets**

|             |        | ACC↑(σ)       | WACC↑(σ)      | MAD↓(σ)       | TAD↓(σ)       | VAR↓(σ)       |
| ----------- | ------ | ------------- | ------------- | ------------- | ------------- | ------------- |
| UCI  Adult  | ERM    | 84.67%(0.58%) | 80.20%(0.82%) | 16.13%(0.82%) | 20.78%(0.99%) | 0.3389(4.77%) |
|             | DRO    | 84.71%(0.26%) | 80.34%(0.36%) | 15.76%(0.35%) | 20.92%(0.24%) | 0.1900(2.05%) |
|             | ARL    | 84.60%(0.63%) | 80.11%(0.91%) | 16.17%(1.05%) | 20.91%(0.95%) | 0.3618(8.41%) |
|             | FairRF | 84.27%(0.13%) | 80.01%(0.15%) | 15.73%(0.18%) | 20.26%(0.58%) | 0.2583(1.38%) |
|             | VFair  | 84.74%(0.34%) | 80.36%(0.49%) | 15.71%(0.73%) | 20.71%(0.80%) | 0.0817(0.98%) |
| Law  School | ERM    | 85.59%(0.67%) | 74.49%(1.84%) | 12.08%(2.74%) | 21.50%(3.35%) | 0.3695(1.37%) |
|             | DRO    | 85.37%(0.88%) | 74.76%(2.08%) | 11.53%(1.91%) | 20.83%(2.37%) | 0.2747(1.43%) |
|             | ARL    | 85.27%(0.71%) | 74.78%(2.12%) | 11.52%(2.21%) | 21.52%(1.97%) | 0.3795(1.80%) |
|             | FairRF | 81.91%(0.27%) | 68.75%(1.61%) | 14.48%(1.65%) | 26.84%(2.20%) | 0.3080(1.59%) |
|             | VFair  | 85.40%(0.99%) | 75.25%(1.51%) | 11.00%(1.92%) | 19.91%(2.43%) | 0.0629(0.24%) |
| COMPAS      | ERM    | 66.70%(0.66%) | 63.20%(1.64%) | 07.15%(1.46%) | 09.12%(1.79%) | 0.1563(3.38%) |
|             | DRO    | 66.37%(0.50%) | 62.41%(1.27%) | 07.51%(1.08%) | 09.58%(1.77%) | 0.1535(2.39%) |
|             | ARL    | 66.65%(0.55%) | 63.27%(1.99%) | 06.93%(1.83%) | 09.09%(3.71%) | 0.1442(3.64%) |
|             | FairRF | 62.90%(0.43%) | 61.55%(1.06%) | 02.64%(1.55%) | 03.69%(2.10%) | 0.0693(1.26%) |
|             | VFair  | 66.80%(0.27%) | 63.86%(0.57%) | 06.25%(0.80%) | 08.47%(1.23%) | 0.0186(0.12%) |
| CelebA      | ERM    | 92.80%        | 89.77%        | 3.64%         | 4.77%         | 0.4008        |
|             | DRO    | 93.04%        | 90.16%        | 3.27%         | 4.74%         | 0.319         |
|             | ARL    | 93.26%        | 89.84%        | 4.02%         | 5.41%         | 0.3738        |
|             | FairRF | -             | -             | -             | -             | -             |
|             | VFair  | 93.43%        | 91.09%        | 2.74%         | 3.85%         | 0.117         |

---

> ### Author Response · Authors · 2023-11-18
>
> **Table 2: Ablation experiments on four benchmark datasets.**
> |             | $\lambda_1$    | $\lambda_2$    | ACC↑(σ)        | WACC↑(σ)      | MAD↓(σ)       | TAD↓(σ)       | VAR↓(σ)       |
> | ----------- | -------------- | -------------- | -------------- | ------------- | ------------- | ------------- | ------------- |
> | UCI  Adult  | $\lambda  = 1$| | 84.71%(0.32%)  | 80.36%(0.44%)  | 15.85%(0.65%) | 20.94%(0.72%) | 0.03(0.32%)   |               |
> |             |                | $\checkmark$   | 84.68%(0.36%)  | 80.27%(0.50%) | 15.91%(0.63%) | 20.97%(0.89%) | 0.0775(0.81%) |
> |             | $\checkmark$ |  | 84.52%(0.44%)  | 80.08%(0.65%)  | 15.99%(0.73%) | 20.92%(0.56%) | 0.0658(1.15%) |               |
> |             | $\checkmark$   | $\checkmark$   | 84.74%(0.34%)  | 80.36%(0.49%) | 15.71%(0.73%) | 20.71%(0.80%) | 0.0817(0.98%) |
> | Law  School | $\lambda  = 1$| | 4.36%(0.11%)   | 74.30%(0.84%)  | 10.88%(0.95%) | 20.73%(1.63%) | 0.0005(0.02%) |               |
> |             |                | $\checkmark$   | 85.4%(0.30%)   | 75.09%(0.58%) | 11.2%(0.82%)  | 20.43%(1.66%) | 0.063(0.14%)  |
> |             | $\checkmark$ |  | 45.39%(28.53%) | 32.03%(14.79%) | 30.31%(3.41%) | 53.12%(5.08%) | 0(0%)         |               |
> |             | $\checkmark$   | $\checkmark$   | 85.40%(0.99%)  | 75.25%(1.51%) | 11.00%(1.92%) | 19.91%(2.43%) | 0.0629(0.24%) |
> | COMPAS      | $\lambda  = 1$ || 55.21%(2.43%)  | 49.9%(3.44%)   | 10.51%(4.34%) | 13.28%(5.51%) | 0(0%)         |               |
> |             |                | $\checkmark$   | 64.29%(0.99%)  | 60.44%(3.63%) | 7.34%(3.76%)  | 9.67%(4.87%)  | 0.0003(0.02%) |
> |             | $\checkmark$ |  | 66.45%(0.85%)  | 63.49%(1.90%)  | 6.6%(2.40%)   | 8.4%(3.12%)   | 0.0191(0.24%) |               |
> |             | $\checkmark$   | $\checkmark$   | 66.80%(0.27%)  | 63.86%(0.57%) | 06.25%(0.80%) | 08.47%(1.23%) | 0.0186(0.12%) |
> | CelebA      | $\lambda  = 1$| | 92.04%         | 89.22%         | 3.66%         | 4.65%         | 12.69%        |               |
> |             |                | $\checkmark$   | 93.46%         | 90.62%        | 3.49%         | 4.67%         | 11.61%        |
> |             | $\checkmark$ |  | 93.23%         | 90.08%         | 3.70%         | 5.14%         | 7.53%         |               |
> |             | $\checkmark$   | $\checkmark$   | 93.43%         | 91.09%        | 2.73%         | 3.85%         | 11.70%        |
>
> **Maintaining superiority consistently under random partitioning.**
>
> Given that our focus is on fairness without demographics accessible during the training stage, our model excels not only in the specific subgroups split by the attributes of "race" and "gender" but also in other possible divisions. Thus, we conducted additional experiments involving over 100 iterations, randomly splitting the data into K groups, ranking the metrics in terms of accuracy and prediction error, and calculating the average rank. For each iteration, we rank the results from the top to the bottom as 1, 2, 3, 4. It's essential to note that FairRF is not included because this method relies on selecting related attributes and is not suitable for this setting. The following results indicate that our method **consistently performs well regardless of the choice of K**.
>
> **Table 3: Average rank of randomly splitting into 4, 10, 20 groups**.
>
> |     |    | K=4  |  |    |      | K=10 |      |      |      | K=20 |  |  |  |
> | ----------- | ----- | ----- | ----- | ----- | ----- | ----- | ----- | ----- | ----- | ----- | ----- | ----- | ----- |
> |     |   | ACC  | WACC | TAD  | MAD  | ACC  | WACC | TAD  | MAD  | ACC  | WACC | TAD  | MAD  |
> | UCI  Adult  | ERM   | 2.5  | 3.3  | 2.93 | 3.01 | 2.5  | 3.33 | 3.25 | 3.19 | 2.5  | 3.42 | 3.13 | 3.29 |
> |    | DRO   | 2.7  | 2    | 2.57 | 2.54 | 2.7  | 2    | 2.33 | 2.29 | 2.7  | 2    | 2.4  | 2.27 |
> |    | ARL   | 2.5  | 3.7  | 3.1  | 3.13 | 2.5  | 3.67 | 3.39 | 3.41 | 2.5  | 3.58 | 3.33 | 3.44 |
> | | VFair | **2.3**  | **1**    | **1.4**  | **1.32** | **2.3**  | **1**    | **1.03** | **1.11** | **2.3**  | **1**  | **1.14** | **1**    |
> | Law  School | ERM   | **1.8**  | 3.46 | 3.04 | 3.04 | **1.8**  | 3.44 | 3.12 | 3.22 | **1.8**  | 3.38 | 3.32 | 3.32 |
> |    | DRO   | 2.7  | 2.02 | 2.68 | 2.71 | 2.7  | 2.09 | 2.61 | 2.48 | 2.7  | 2.18 | 2.4  | 2.33 |
> |     | ARL   | 2.8  | 3.52 | 3.03 | 3.12 | 2.8  | 3.47 | 3.2  | 3.26 | 2.8  | 3.38 | 3.28 | 3.35 |
> |   | VFair | 2.7  | **1**    |**1.25** |**1.13** | 2.7  | **1**   | **1.07** | **1.04** | 2.7  | **1.06** | **1**    | **1**    |
> | COMPAS      | ERM   | 2.6  | 3.02 | 3.05 | 3.09 | 2.6  | 3.08 | 2.87 | 2.94 | 2.6  | 2.67 | 3.03 | 3.17 |
> |     | DRO   | 2.8  | 3.21 | 3.08 | 3    | 2.8  | 3.14 | 3.17 | 3.07 | 2.8  | 2.88 | 3.15 | 3.09 |
> |    | ARL   | 2.5  | 2.77 | 2.83 | 2.88 | 2.5  | 2.71 | 2.92 | 2.99 | 2.5  | 2.53 | 2.82 | 2.74 |
> |    | VFair | **2.1**  | **1**    |**1.04** | **1.03** | **2.1**  | **1.07** | **1.04** | **1** | **2.1**  | **1.92** | **1**    | **1**    |

---

> > ### Author Response · Authors · 2023-11-22
> >
> > **Comparison with fairness methods with accessible sensitive attributes.**
> >
> > We have further supplemented control experiment, where the model has access to sensitive attributes and is optimized under constrained regularization. In detail, we reproduced the MMPF in [1] and further designed experiments that penalize the losses of the minority group, denoted as PMG. Take UCI Adult and Law School as two examples, the results are presented in Table 4.
> >
> > **Table 4: Comparison with methods with access to attributes.**
> >
> > |            |       | ACC↑   | WACC↑  | TAD↓   | MAD↓   | VAR↓   |
> > | ---------- | ----- | ------ | ------ | ------ | ------ | ------ |
> > | UCI Adult  | PMG   | 79.48% | 73.02% | 22.36% | 27.85% | 4.1686 |
> > |            | MMPF  | 85.19% | 81.09% | 12.73% | 17.73% | -      |
> > |            | VFair | 84.74% | 80.36% | 15.71% | 20.71% | 0.0817 |
> > | Law School | PMG   | 78.52% | 70.14% | 9.47%  | 15.59% | 0.8183 |
> > |            | MMPF  | 82.85% | 76.20% | 8.42%  | 15.43% | -      |
> > |            | VFair | 85.40% | 75.25% | 11.00% | 19.91% | 0.0629 |
> >
> > By leveraging additional group information, MMPF achieves improved results. However, MMPF is not a harmless approach, particularly evident on Law School, where it sacrifices model utility for a fairer point. PMG yields unsatisfactory performance due to its excessive focus on the minority group, missing general information from other groups.
> >
> > **Comparison of F1-score on CelebA.**
> >
> > On CelebA, which is observed as an imbalanced multi-classification task, we further compare the F1-score of each method. Similar to the ACC-based metrics, we denote F, WF, MFD, and TFD as F1-score, Worst F1-score, Max F1-score Disparity, and Total F1-score Disparity, respectively. As shown in Table 5, our model significantly outperforms others, resulting in better utility and better fairness. In detail, the imbalanced multiclassification task setting makes the fairness problem more challenging. When evaluated under the F1-score, which considers the imbalance of each label, the worst-performing groups exhibit a larger gap with the mainstream groups, providing our method with more opportunity to address these disparities.
> >
> > **Table 5:  Comparison of F1-score on CelebA.**
> >
> > |       | F↑     | WF↑    | TFD↓   | MFD↓   |
> > | ----- | ------ | ------ | ------ | ------ |
> > | ERM   | 91.40% | 70.17% | 19.39% | 22.82% |
> > | DRO   | 91.57% | 70.16% | 20.68% | 25.25% |
> > | ARL   | 91.60% | 70.39% | 20.14% | 24.33% |
> > | VFair | **91.91%** | **75.70%** | **14.39%** | **18.50%** |
> >
> > [1] BERTRAN M, MARTINEZ N, SAPIRO G. Minimax Pareto Fairness: A Multi Objective Perspective[J]. International Conference on Machine Learning, International Conference on Machine Learning, 2020.

---

### Meta-Review · Area_Chair_tXev · 2023-12-09

**Metareview:**

The authors address the challenge of training a fair model with comparable accuracy across all protected groups, where the model is unaware of the group membership during training. The central concept of the paper revolves around achieving low loss for each observation during training. This involves minimizing both the expected loss across observations and the variance of the loss. The authors formalize this update as a Lagrangian and propose heuristics for updating in a "harmless" manner, allowing the variance to decrease without significantly impacting the expectation. Through experiments on four diverse datasets, the authors demonstrate that their method achieves accuracy on par with other approaches while attaining lower variance and improved fairness. Ablation experiments further support the effectiveness of their heuristic updates in enhancing performance.

Reviewers mostly concerned about 1) the empirical improvements from method seems marginal and incremental, and sometimes indistinguishable from the other baselines; 2) the overlap between the presented idea with several works that leverage variance minimization (that outside of the DRO framework).

**Justification For Why Not Higher Score:**

The reviewers were concerned about the overlap between the idea and several existing works that adopted similar variance control idea. The authors are also encouraged to revisit the experiments - so far the improvements are insufficient to justify the merits of the proposed solution.

**Justification For Why Not Lower Score:**

N/A

---

### Decision · Program_Chairs · 2024-01-16

Reject